# BEYOND SCALING LAWS: UNDERSTANDING TRANSFORMER PERFORMANCE WITH ASSOCIATIVE MEMORY

## ABSTRACT

Increasing the size of a Transformer does not always lead to enhanced performance. This phenomenon cannot be explained by the empirical scaling laws. Furthermore, the model's enhanced performance is closely associated with its memorization of the training samples. We present a theoretical framework that sheds light on the memorization during pre-training of transformer-based language models. We model the behavior of Transformers with associative memories using Hopfield networks, such that each transformer block effectively conducts an approximate nearest-neighbor search. Based on this, we use a distance-based energy function to approximate the one in the modern continuous Hopfield network, which provides an insightful explanation for the attention mechanism. Since the softmax function corresponds to the gradient of the LogSumExp function in the energy, using the majorization-minimization technique, we construct a global energy function to capture the layered architecture. We show a dependency between the model size and the dataset for the model to attain optimal performance, and the achievable cross-entropy loss is bounded below.

## 1 INTRODUCTION

Transformer-based neural networks have exhibited powerful capabilities in accomplishing a myriad of tasks such as text generation, editing, and question-answering. These models are rooted in the Transformer architecture (Vaswani et al., 2017) which employs the self-attention mechanisms to capture the context in which words appear, resulting in superior ability to handle long-range dependencies and improved training efficiency. In many cases, models with more parameters result in better performance measured by perplexity (Kaplan et al., 2020), as well as in the accuracies of end tasks (Khandelwal et al., 2019; Rae et al., 2021; Chowdhery et al., 2023). As a result, larger and larger models are being developed in the industry. Recent models (Smith et al., 2022) can reach up to 530 billion parameters, trained on hundreds of billions of tokens with more than 10K GPUs.

Nevertheless, it is not always the case that bigger models result in better performance. For example, the 2B model MiniCPM (Hu et al., 2024c) exhibits comparable capabilities to larger language models, such as Llama2-7B (Touvron et al., 2023), Mistral-7B (Jiang et al., 2023), Gemma-7B (Banks & Warkentin, 2024), and Llama-13B (Touvron et al., 2023). Moreover, as computational resources for training larger models increase, the size of available high-quality data may not keep pace. It has been documented that the generalization abilities of a range of models increase with the number of parameters and decrease when the number of training samples increases (Belkin et al., 2019; Nakkiran et al., 2021; d'Ascoli et al., 2020), indicating that generalization occurs beyond the memorization of training samples in over-parameterized neural networks (Power et al., 2022). Therefore, it is crucial to understand the convergence dynamics of training loss during memorization, both in relation to the model size and the dataset at hand. There has been an increasing interest in the empirical scaling laws under constraints on the training dataset size (Muennighoff et al., 2024). Extensive experiments have led to the conclusion of the empirical scaling laws (Kaplan et al., 2020) in terms of the model performance measured by the test cross-entropy loss. Unfortunately, this scaling law does not explain why in many cases smaller models perform better.

In this paper, we focus on the theoretical aspects of the dependencies between the achievable performance, indicated by the pre-training loss, for transformer-based models, and the model and data sizes during memorization. It has been observed that a family of large language models tends

to rely on knowledge memorized during training (Hsia et al., 2024), and the larger the models, the more they tend to encode the training data and organize the memory according to the similarity of textual context (Carlini et al., 2022; Tirumala et al., 2022). Therefore, we model the behavior of the Transformer layers with associative memory, which associates an input with a stored pattern, and inference aims to retrieve the related memories. A model for associative memory, known as the Hopfield network, was originally developed to retrieve stored binary-valued patterns based on part of the content (Amari, 1972; Hopfield, 1982). Recently, the Modern Continuous Hopfield Network (MCHN) was proposed and has been shown to exhibit equivalence to the attention mechanism (Ramsauer et al., 2020). However, the MCHN only explains an individual Transformer layer and relies heavily on regularization.

Transformer-based models consist of a stack of homogeneous layers. The attention and feed-forward layers contribute to the majority of the parameters in large models and are also the key components of the attention mechanism. Furthermore, the layered structure of the transformer networks induces a sequential optimization, reminiscent of the majorization-minimization (MM) technique (Ortega & Rheinboldt, 1970; Sun et al., 2016), which has been extensively utilized across domains such as signal processing and machine learning. Using the MM framework, we construct a global energy function tailored for the layered structure of the transformer network.

Our model provides a theoretical framework for analyzing the performance of transformer-based language models as they memorize training samples. Large language models only manifest capabilities for certain downstream tasks once the training loss reaches a specific threshold (Du et al., 2024). In practice, the training of large language models is terminated when the loss curves plateau. On the one hand, the validation loss offers valuable insights for budgetary considerations; it has been observed that even after training on up to 2T tokens, some models have yet to exhibit signs of saturation (Touvron et al., 2023). On the other hand, implementing early stopping can potentially compromise the generalization capabilities of the models (Murty et al., 2023). In Appendix F, we include a series of experiments utilizing GPT-2, vanilla Transformer, and OpenELM models on various data. The experimental outcomes provide evidence to support our theoretical results. We believe this work offers valuable theoretical perspectives on the pre-training of large language models.

**Our Contribution:** (1) We take a new perspective by studying Transformer behavior using associative memories with Hopfield networks. We reveal the underlying connection between the attention mechanism and nearest-neighbor search. (2) We approximate the continuous Hopfield network using a distance-based energy function, excluding additional regularization terms. By recognizing that the softmax function corresponds to the gradient of the LogSumExp function, we employ the majorization-minimization technique to construct a global energy function to accommodate the layered architecture of the Transformer. (3) Using our theoretical framework, we characterize the dependencies between pre-training loss, model size, and dataset during memorization for transformer-based language models.

## 2 RELATED WORK

**Scaling laws.** Empirical evidence suggests that the performance of models increases as both the size of the models and the volume of training data scale up (Kaplan et al., 2020; Khandelwal et al., 2019; Rae et al., 2021; Chowdhery et al., 2023). Intensive experiments on transformer-based large language models have also been conducted to explore neural scaling laws under various conditions, including constraints on computational budget (Hoffmann et al., 2022b), data (Muennighoff et al., 2024), and instances of over-training (Gadre et al., 2024). In these analyses, a decomposition of the expected risk is utilized, leading to the following fit:

$$\hat{L}(N, D) = E + \frac{A}{N^\alpha} + \frac{B}{D^\beta}, \tag{1}$$

where $N$ and $D$ denote the number of parameters of the model and the size of the training data respectively. For Chinchilla models, the fitted parameters are (Hoffmann et al., 2022a)

$$\alpha = 0.34, \quad \beta = 0.28, \quad E = 1.61, \quad A = 406.4, \quad B = 410.7.$$

A line of research concerns the generalization of over-parameterized neural networks (Belkin et al., 2019; Nakkiran et al., 2021; Power et al., 2022). Recent experiments show that over-trained Transformers exhibit inverted U-shaped scaling behavior (Murty et al., 2023), which is not explained

by the empirical scaling laws. Further discussions on the relationship between our method and the Chinchilla scaling laws are deferred to Appendix A.

**Energy-based models.** Energy-based models (LeCun et al., 2006), motivated by statistical physics, have become a fundamental modeling tool in various fields of machine learning over the past few decades. The central idea is to model the neural network through a parameterized probability density function $p_\theta(x)$ for $x \in \mathbb{R}^n$ and to express the distribution in terms of a learnable energy function $E_\theta(x) : \mathbb{R}^n \mapsto \mathbb{R}$ whose parameters correspond to the model's parameters as $p_\theta(x) = \frac{\exp(-E_\theta(x))}{Z_\theta}$. Here, $Z_\theta = \int \exp(-E_\theta(x)) \, \mathrm{d}x$ is the normalizing constant known as the partition function.

**Hopfield models.** Classical Hopfield networks (Amari, 1972; Hopfield, 1982) were introduced as paradigmatic examples of associative memory. The network's update dynamics define an energy function, whose fixed points correspond to the stored memories. An important indicator is the number of patterns that the model can memorize, known as the network's storage capacity. Modifications to the energy function (Krotov & Hopfield, 2016; Demircigil et al., 2017) result in higher storage capacities (see Table 1 in Appendix B). The original model operates on binary variables, and continuous Hopfield Networks have been developed later (Hopfield, 1984). The modern continuous Hopfield network (MCHN) (Ramsauer et al., 2020) connects the continuous formulation with the attention mechanism by introducing a specific model with a softmax activation function. Given an input (e.g., a prompt), the Hopfield layer retrieves a memory by converging to a local minimum of the energy landscape, and the update rule has a nice correspondence to the query-key-value mechanism in attention. Krotov (2021) proposes a Hierarchical Associative Memory (HAM) model with a global energy function for layered networks, as opposed to energy functions for individual layers. Further discussions on the relationship between the energy function utilized in this paper and other energy functions found in the literature on Hopfield networks is detailed in Appendix A.

## 3 SYSTEM MODEL

We consider tokenized training samples $\mathcal{D} = \{s^1, s^2, \ldots, s^d\}$, where each element is a sequence of tokens whose length is bounded by a number $T_{\max} \in \mathbb{N}$. Let $\widetilde{\mathcal{D}} = \{\tilde{s}^1, \tilde{s}^2, \ldots, \tilde{s}^{d'}\}$ be the set of held-out validation samples. Details on the pre-processing of the dataset is described in Appendix E. The size $D \in \mathbb{N}$ of the dataset is proportional to the number of samples $d \in \mathbb{N}$. Let $d_{\mathrm{emb}} \in \mathbb{N}$ be the embedding dimension of the tokens, so each input sequence has $n = T_{\max} d_{\mathrm{emb}}$ dimensions. Let $N \in \mathbb{N}$ be the number of parameters in the attention layers and the feed-forward layers, which constitute most of the parameters in the Transformer model. Suppose there are $l$ layers, then

$$N \approx A l d_{\mathrm{emb}}^2 = \frac{A l d_{\mathrm{emb}}}{T_{\max}} n, \qquad D \approx T_{\max} d. \tag{2}$$

for some constant $A$ (see Appendix E). We use a generic distance metric $d(\cdot, \cdot)$ in the Euclidean space. In practice, this metric can usually be particularized to be the Euclidean norm.

### 3.1 ASSOCIATIVE MEMORIES

We consider models trained with a causal language modeling objective. Given an input sequence of tokens $s_{\leq t} = (s_1, s_2, \ldots, s_t)$, the $l$-layer Transformer outputs a distribution over the next token $s_{t+1}$. The transformer models are trained to maximize the log-probability of the correct token $s_{t+1}$ given $s_{\leq t}$. During pre-training, input texts are segmented into sequences of length $T_{\max}$, and the model develops the ability to generate desired content, such as predicting the correct continuations. Thus, the sequences can be viewed as patterns in the setting of *associative memories*, where stored *patterns* (e.g., sequences) can be retrieved using partial contents of the patterns. Since each prediction may have access to a varying number of preceding tokens, the sequences padded to a length of $T_{\max}$ earlier may have less amount of information for the associative memory retrieval. As demonstrated in (Svete & Cotterell, 2024; Svete et al., 2024), a Transformer can be modeled as a representation-based $n$-gram model with relative small $n$. Consequently, only a small number of padded sequences are affected by this discrepancy due to non-uniform lengths, and it does not significantly influence the collective behavior as analyzed within the framework of statistical physics. We note that, for objectives that are

not causal, such as masked token modeling and sequence-to-sequence modeling, analogous reasoning can be applied within the latent space.

It has been observed that the models tend to memorize patterns from the training data (Carlini et al., 2021; Biderman et al., 2024). Empirical studies on large language models have shown that the larger the models are, the more they tend to memorize training data (Carlini et al., 2022; Tirumala et al., 2022). This memorization allows the models to learn important patterns, such as world knowledge (Hsia et al., 2024), individual words (Chang & Bergen, 2022), and linguistic structure (Chang & Bergen, 2024). In light of these findings, we make the following assumption regarding memorization. As passing the text data through an embedding layer reduces their correlation, we posit that the Transformer blocks serve to store the resulting latent representations, which are extracted from the sequences once they are embedded.

**Assumption 1.** *During the pre-training process, the model memorizes the (latent) training samples* $\mathcal{D}$ *as patterns* $\{\rho^1, \rho^2, \ldots, \rho^d\}$, *where* $\rho^i \in \mathbb{R}^n$ *for* $i = 1, 2, \ldots, d$.

To be economical with notations, we use $\mathcal{D}$ to directly address the patterns $\mathcal{D} = \{\rho^1, \rho^2, \ldots, \rho^d\}$. By memorizing the samples, we mean that the patterns are stored within the model and can be retrieved when provided with an adequate prompt. Specifically, we follow the definitions in (Ramsauer et al., 2020) for pattern storage and retrieval.

**Definition 1.** *For every pattern* $\rho^i$, *denote by* $B_i := \{x \in \mathbb{R}^n : d(x, c_i) \leq r_i\}$ *an* $n$-*ball such that* $\rho^i \in B_i$. *The pattern* $\rho^i$ *is said to be* ***stored*** *if there exists a single fixed point* $\rho^{i*} \in B_i$ *to which all points* $x \in B_i$ *converge, and* $B_i \cap B_j = \emptyset$ *for* $i \neq j$. *Such* $B_i$ *is said to be associated to the pattern* $\rho^i$, *and we denote* $B_i \sim \rho^i$. *The pattern* $\rho^i$ *is said to be* ***retrieved*** *if the converged point is* $\epsilon$-*close to the fixed point* $\rho^{i*}$.

**Remark 1.** *In the work of Saha et al. (2023), a collective attraction mechanism is introduced to address the challenge of partial cluster assignments using associative memory. When the intersection of clusters* $B_i$ *and* $B_j$ *is non-empty* $(B_i \cap B_j \neq \emptyset)$, *analogous relaxations to those presented in Equation (7) of (Saha et al., 2023) can be made.*

Without loss of generality, we assume $c_i = \rho^i$. We also assume that the small set of held-out test samples exhibits the same patterns as those in the training set. In practice, the validation samples are randomly selected from the same dataset as the training samples, preserving the distribution.

**Assumption 2.** *We posit that the latent representations derived from the validation set, after being processed through an embedding layer, are stored in a manner analogous to those extracted from the training set. Specifically, for every element in the set of latent patterns* $\widetilde{\mathcal{D}}$ *corresponding to the validation data, there exists an* $B_i$ *for some* $i \in [d]$. *Consequently, we assume that* $\widetilde{\mathcal{D}} \subset \mathcal{D}$.

### 3.2 TRANSFORMER BLOCKS

Transformer-based models, originated by Vaswani et al. (2017), are often made of a stack of homogeneous layers. The multi-head attention and feed-forward (FF) layers account for most of the parameters in the model. Appendix E provides more details using GPT-2 as an example.

**Attention mechanism.** The attention mechanism arguably contributes most to the overall performance of the transformer models. The attention mechanism takes three matrices $W_K \in \mathbb{R}^{d_{\text{emb}} \times d_k}$, $W_Q \in \mathbb{R}^{d_{\text{emb}} \times d_k}$, and $W_V \in \mathbb{R}^{d_{\text{emb}} \times d_v}$ as weights that can be interpreted as *keys, queries,* and *values*. Setting $d_v = d_{\text{emb}}$ facilitates the inclusion of residual connections. In a single update, an attention matrix is obtained using the update rule $\text{Attention}(Q, K, V) = V \cdot \text{softmax}(QK^{\mathsf{T}}/\sqrt{d_k})$.

**Feed-forward layers.** It has been shown that the FF layers operate essentially as key-value memories (Geva et al., 2020) such that $\text{FF}(x) = f(x \cdot K^{\mathsf{T}}) \cdot V$, where $K, V$ are parameter matrices and $f$ is a non-linear activation function such as ReLU. The FF layers can be merged into the attention without degrading the Transformer's performance (Sukhbaatar et al., 2019). Thus, the attention layer and the FF layer can be conceptually integrated into a unified transformer layer.

As the model scales, the attention and FF layers, being stacked, constitute the majority of the model's parameters. Also, since the fundamental operations of the Transformer are the attention and FF layers, we consider the number of parameters $N$ in these layers, which is almost proportional to

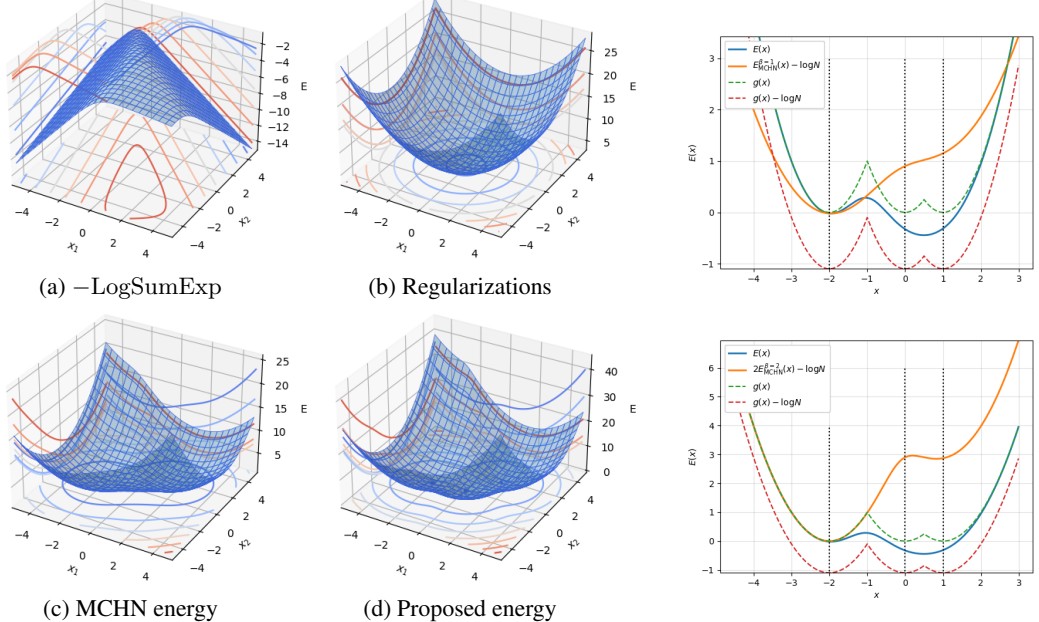

(a) $-$LogSumExp  (b) Regularizations

(c) MCHN energy  (d) Proposed energy

Figure 1: **Left:** *Energy landscapes for a set of 2-dimensional patterns* $\mathcal{D} = \{(-2, -0.5), (0.2, -0.3), (1.5, 1.5)\}$. (a) The negative LogSumExp function with $\beta = 1$, as an extension of (Demircigil et al., 2017). (b) The regularization terms $\frac{1}{2}x^T x + \beta^{-1} \log d + \frac{\max_i \|\rho^i\|^2}{2}$ in the MCHN energy. (c) The MCHN energy $E_{\text{MCHN}}^1(x)$. (d) The layer-wise energy equation 4 with squared Euclidean norm. **Right:** *Energy landscapes for a set of 1-dimensional patterns* $\mathcal{D} = \{-2, 0, 1\}$. The orange curves correspond to the MCHN energy with $\beta = 1, 2$.

the square of the embedding dimension. The ratio depends on the number of layers and the hidden dimensions of the transformer blocks. In the current work, we do not consider other modifications such as lateral connections, skip-layer connections, mixture of experts, mixture of depths, routing, or other compressive modules such as (Xiong et al., 2023; Fei et al., 2023; Munkhdalai et al., 2024).

## 4 A GLOBAL ENERGY FUNCTION

For the attention layer, we employ an energy function that does not rely on additional regularization terms based on a distance metric. We then adapt this function to the layered transformer blocks using the majorization-minimization technique. For reference, related energy functions for Hopfield networks are listed in Table 1 in Appendix B. In particular, let $M := (\rho^1, \rho^2, \ldots, \rho^d)$, the energy function for the modern continuous Hopfield network (Ramsauer et al., 2020) is

$$E_{\text{MCHN}}^\beta(x) = -\text{LogSumExp}(\beta, M^\mathsf{T} x) + \frac{1}{2}x^\mathsf{T} x + \beta^{-1} \log d + \frac{\max_i \|\rho^i\|^2}{2}, \quad \text{where}$$

$$\text{LogSumExp}(\beta, y) := \beta^{-1} \log \left( \sum_{i=1}^d \exp(\beta y_i) \right), \quad x \in \mathbb{R}^n, \quad y = (y_1, \ldots, y_d) \in \mathbb{R}^d.$$

It can be readily observed that the negative LogSumExp function was adapted from (Demircigil et al., 2017). However, in the continuous domain, the negative LogSumExp function is not convex, making it a less suitable candidate for the energy function. The MCHN energy then adds regularization terms to create a convex energy function. These regularization terms involve both the max norm of the input and the number of patterns.

### 4.1 A LAYER-WISE ENERGY FUNCTION

Instead of designing different regularization terms, we apply an energy function by considering an auxiliary function

$$g(x) := \min_{1 \le i \le d} d(x, \rho^i), \tag{3}$$

which corresponds to the *nearest neighbor search* over the the set of patterns $\mathcal{D}$. So it holds that $g(x) \geq 0$, with $g(x) = 0$ if and only if $x \in \mathcal{D} = \{\rho_1, \ldots, \rho_d\}$. According to Assumption 1, the model has memorized the patterns through pre-training; thus, the inference corresponds to a search algorithm based on some distance $d(\cdot, \cdot)$. We use the squared Euclidean 2-norm $d(x, y) = \|x - y\|^2$ in the sequel. We consider a function $E(x)$ which also takes the form of LogSumExp.

$$E(x) = -\log \left( \sum_{i=1}^{d} \exp(-d(x, \rho^i)) \right). \tag{4}$$

It is worth noting that the softmax function is the gradient of the LogSumExp function. So the Transformer integrates the search over layers. By summing up the negative distance between $x$ and each stored pattern, the function assigns smaller values to points closer to the patterns. The distance-based energy function equation 4 with an inverse temperature has been applied by Saha et al. (2023) within the context of clustering. This energy function is well-suited to a more generalized framework, namely the universal Hopfield networks (Millidge et al., 2022). By replacing the dot product in the MCHN energy with the distance metric, $E(x)$ achieves similar goal without additional regularization. As shown in Figures 1a and 1b, as an extension of (Demircigil et al., 2017), the negative LogSumExp is not convex in the real domain, so regularization terms are applied in MCHN. Figures 1d and 1c show that the landscape of the proposed energy resembles that of the MCHN energy. In (Ramsauer et al., 2020), it is shown that $E_{\text{MCHN}}$ induces stationary points near the stored patterns. Here, the energy function $E(x)$ serves as a smooth surrogate of the desired function $g(x)$ in equation 3, therefore also demonstrates the retrieval ability.

**Proposition 1.** *Given* $\mathcal{D} = \{\rho_1 \ldots, \rho_d\}$, *the layer-wise energy* $E(x)$ *satisfies*

$$g(x) - \log d \leq E(x) \leq g(x).$$

The proof of Proposition 1 is due to Lemma 3 and is deferred to Appendix C. Furthermore, we show that $E(x)$ is close to the MCHN energy, as delineated below.

**Proposition 2.** *Let* $\beta = 2$ *we have*

$$|E(x) - (2E_{\text{MCHN}}^{\beta=2}(x) - \log d)| \leq \max_{1 \leq i \leq d} \|\rho^i\|^2 - \min_{1 \leq i \leq d} \|\rho^i\|^2.$$

The proof is given in Appendix C. Fig. 1 provides visualizations for the two propositions using low dimensional patterns. The following result follows directly from the above inequalities.

**Proposition 3.**

$$\min_{1 \leq i \leq d} \|\rho^i\|^2 - \max_{1 \leq i \leq d} \|\rho^i\|^2 \leq g(x) - 2E_{\text{MCHN}}^{\beta=2}(x) \leq \max_{1 \leq i \leq d} \|\rho^i\|^2 - \min_{1 \leq i \leq d} \|\rho^i\|^2 + \log d.$$

Since the energy function equation 4 and the MCHN energy both approximate the search for the nearest pattern (desired stationary point), according to Theorem 4 in (Ramsauer et al., 2020), in each transformer layer, the probability density of the transformer layer, corresponding to the retrieval, is $p(x) = \frac{1}{Z} \exp(-E(x)|_\Omega)$, where $Z$ is the normalizing factor, $\Omega = \bigcup_{i=1}^{d} B_i$, and $B_i$ is as defined in Definition 1. We assume that $B_i$ is centered at the $i$-th pattern. We make the following assumption on the samples, such that the (latent) patterns are well-separated.

**Assumption 3.** *Passing the input through an embedding layer reduces the correlation between the original samples. Therefore the patterns in the latent space in* $\mathcal{D}$ *are well-separated, i.e.,* $B_i \cap B_j = \emptyset, \forall 1 \leq i < j \leq d.$

Under Assumption 3, the energy function, confined in $\Omega$, can be replaced by the nearest neighbor search $g(x)$. So the probability density is

$$p(x) = \frac{1}{Z} \exp(-g(x)). \tag{5}$$

### 4.2 THE LAYERED STRUCTURE

As discussed in the related works, most Hopfield models only handle a single hidden layer, whereas SoTA transformer-based models often consist of a stack of homogeneous blocks of attention and

FF layers. To model the multi-layered structure of Transformers, we employ a technique known as majorization-minimization (MM) (Ortega & Rheinboldt, 1970; Sun et al., 2016), which aims to accelerate optimization using surrogate convex functions. We argue that the layered structure serves the same purpose when the patterns memorized by all layers encompass the set of learned representations.

We divide the set of samples into $\mathcal{D} = \cup_{i=1}^{l} \mathcal{D}_i$, where $\mathcal{D}_i = \{\rho^{i_1}, \rho^{i_2}, \ldots, \rho^{i_{d_i}}\}$. Then, the energy function for each layer can be written as

$$E_t(x) = \frac{1}{Z_t} \exp(-g_t(x)), \quad \text{where} \quad g_t(x) := \min_{1 \le j \le d_t} d(x, \rho^{t_j}).$$

Denote by $x^{(0)}$ the embedding vector input into the first transformer layer and $x^{(t)} \in \mathbb{R}^n$ the output of the $t$-th layer for $t = 1, 2, \ldots, l$. Let $E_t(x)$ be the energy function associated with the Hopfield model of the $t$-th layer, then the sequential structure of the transformer network is achieved by forwarding the output $x^{(t-1)}$ to the $t$-th layer as input, i.e.,

$$x^{(t)} = \arg\min_{x \in \mathcal{X}_t} E_t(x), \quad \mathcal{X}_t = \{x \in \mathbb{R}^n : d(x, x^{(t-1)}) \le \delta_t\}, \qquad t = 1, \ldots, l \qquad (6)$$

where the retrieved fix point attractor in the $t$-th layer is $\delta_t$-close to $x^{(t-1)}$ in $d(\cdot, \cdot)$ for some $\delta_t > 0$. Such sequential optimization step is equivalent to the MM technique where every minimization step locally approximates the objective function. In particular, equation 6 corresponds to the surrogate function Eq. (3) in (Sun et al., 2016). Therefore, we define a global energy function

$$E_{\text{global}}(x) := -\text{LogSumExp}((-E_1(x), -E_2(x), \ldots, -E_l(x))). \qquad (7)$$

$E_{\text{global}}(x)$ is continuous but not convex. As opposed to the HAM (Krotov, 2021), the global energy function is not a linear combination of the component energies. According to Lemma 3, we have

$$\min_{1 \le i \le l} E_i(x) - \log l \le E_{\text{global}}(x) < \min_{1 \le i \le l} E_i(x). \qquad (8)$$

So $E_t(x)|_{x \in \mathcal{X}_t} \ge E_{\text{global}}(x)|_{x \in \mathcal{X}_t} + c_t$ as in Eq. (2) in (Sun et al., 2016). The probability density function corresponding to the layered transformer network can then be written as

$$p_\theta(x) = \frac{1}{Z_\theta} \exp(-E_{\text{global}}(x)), \quad x \in \Omega \qquad (9)$$

where $\theta$ denotes the model's parameters and $Z_\theta$ is the normalizing constant.

## 5 CROSS-ENTROPY LOSS

We now proceed to analyze the cross-entropy loss, a metric that quantifies the divergence between predicted probabilities and actual labels, and is widely utilized for training Transformer models.

### 5.1 A LOWER BOUND

The attention mechanism encompasses a softmax operation that generates a probability distribution $p \in \Delta_n$. In practice, the final softmax output is subsequently input into a task-specific layer to facilitate downstream tasks, such as predictions and classifications. The attention softmax influences the model's ability to understand and process the input data, which in turn affects the output probabilities that are used to calculate the cross-entropy loss. In essence, the attention softmax indirectly influences the cross-entropy loss by shaping the model's predictions. Consequently, we evaluate the alignment between the final softmax output of the transformer blocks and the target distribution. We demonstrate that the cross-entropy loss can be articulated through the logarithm of the partition function of the model's distribution. This formulation reveals how the allocation of attention weights is contingent upon the learned patterns, thereby establishing a relationship between the characteristics of the training data and the model size, which is pivotal for attaining optimal performance.

Let us consider the cross-entropy loss on the validation set $\widetilde{\mathcal{D}}$. Generally, the cross-entropy loss is the negative log-likelihood computed over a mini-batch. Since we are considering un-batched validation

samples, the loss is normalized by the size $d'$. According to equation 8, there exist a layer $t$ such that $E_{\text{global}}(x)$ is close to $E_t(x)$, i.e.,

$$E_{\text{global}}(x) = E_t(x) - \log l + c(x), \qquad c(x) \in C^\infty(\mathbb{R}^n) \tag{10}$$

such that $0 \le c(x) < \log l$. To simplify, we further assume that $c(x) = c \in [0, \log l)$ is constant. Under Assumption 1, the target distribution, which encodes all the patterns in $\mathcal{D}$, is given by

$$p_{\mathcal{D}}(x) = \sum_{i=1}^{d} p_i \delta(x - \rho^i), \quad x \in \mathbb{R}^n$$

where $\delta(\cdot)$ is the Dirac delta function such that $\delta(x) = 0, \forall x \ne 0$ and $p_i = \Pr(x = \rho^i)$ is the probability mass assigned to pattern $\rho^i$ for $i = 1, 2, \ldots, d$. Suppose the data points are homogeneous, i.e., $p_i = \frac{1}{d}$, then $P_{\mathcal{D}}(x) = \frac{1}{d} \sum_{i=1}^{d} \delta(x - \rho^i)$, and the corresponding test samples $\widetilde{\mathcal{D}}$ induces

$$P_{\widetilde{\mathcal{D}}}(x) = \frac{1}{d'} \sum_{i=1}^{d'} \delta(x - \rho^{\sigma(i)}), \quad \sigma(\cdot) \in \text{Sym}([d]). \tag{11}$$

**Proposition 4.** *Let $L$ be the cross-entropy loss of the above model, then*

$$L \approx \log Z_t + \frac{1}{Z_t} \ge 1, \qquad \text{where } c \in [0, \log l).$$

The proof is deferred to Appendix D.1. Note that the empirically obtained loss function equation 1 for the Chinchilla model converges to $\hat{L}(N, D) = 1.61$ as $N \to \infty$ and $D \to \infty$, which corroborates our theory that $L(N, D) \approx \log Z_t + \frac{1}{Z_t} \ge 1$, with minimum obtained when $Z_t = 1$.

## 5.2 INTERDEPENDENCY BETWEEN MODEL SIZE AND TRAINING DATA

Next, we explore the optimal balance between model size and data during memorization. Let $B_t(x)$ denote the $n$-ball with radius $t$ centered at $x$. Let $A_{n-1} = \frac{2\pi^{n/2}}{\Gamma(\frac{n}{2})}$ represent the hyper-volume of the $(n-1)$-dimensional unit sphere.

$$\gamma(n, r) = \int_0^r t^{n-1} e^{-t} \, \mathrm{d}t, \quad \Gamma(n, r) = \int_r^\infty t^{n-1} e^{-t} \, \mathrm{d}t$$

are the lower and upper incomplete gamma functions. Then

$$\int_{x \in B_i} \exp(-\|x - \rho^i\|^2) \, \mathrm{d}x = \int_{\|x - \rho^i\| < r_i} \exp(-\|x - \rho^i\|^2) \, \mathrm{d}x = \int_{\|y\| < r_i} \exp(-\|y\|^2) \, \mathrm{d}y$$

$$= \int_0^{r_i} \int_{\partial B_t(0)} e^{-t^2} \, \mathrm{d}\mathcal{H}^{n-1} \mathrm{d}t = \int_0^{r_i} e^{-t^2} \mathcal{H}^{n-1}(\partial B_t) \mathrm{d}t \tag{12}$$

$$= \int_0^{r^i} e^{-t^2} A_{n-1} t^{n-1} \, \mathrm{d}t = \frac{2\pi^{\frac{n}{2}}}{\Gamma(\frac{n}{2})} \int_0^{r_i} t^{n-1} e^{-t^2} \, \mathrm{d}t = 2\pi^{\frac{n}{2}} \frac{\gamma(n, r_i)}{\Gamma(\frac{n}{2})}.$$

We take a closer look at the layer partition function, which gives us

$$Z_t = \int_{x \in \Omega} \exp(-g_t(x)) \, \mathrm{d}\mu = \int_{x \in \Omega} \exp(-\min_i d(x, \rho^{t_i})) \, \mathrm{d}\mu$$

$$= \sum_{i=1}^{d} \int_{x \in B_{t_i}} \exp(-\|x - \rho^{t_i}\|^2) \, \mathrm{d}x \stackrel{(12)}{=} 2 \sum_{i=1}^{d} \pi^{\frac{n}{2}} \frac{\gamma(n, r_i)}{\Gamma(\frac{n}{2})}, \tag{13}$$

where $r_i$ is the radius of $B_i$. According to Lemma 5, we have

$$e^{-r_i} V_n(r_i) \le 2\pi^{\frac{n}{2}} \frac{\gamma(n, r_i)}{\Gamma(\frac{n}{2})} = \int_{x \in B_i} \exp(-\|x - \rho^i\|^2) \, \mathrm{d}x \le V_n(r_i),$$

where $V_n(r) = \pi^{\frac{n}{2}} r^n / \Gamma(1 + \frac{n}{2})$ is the hyper-volume of the $n$-dimensional ball of radius $r$. Note that the volume of the unit ball $V_n(1)$ in higher dimensions decreases fast with respect to the increase

in dimensionality. The stability of the probabilities is attributed to the application of normalization operators, including LayerNorm (Xiong et al., 2020) and RMSNorm (Zhang & Sennrich, 2019), which regulate the distribution of activations. The gamma function can be approximated using Stirling's approximation (Appendix D.2) for large values of its argument, which gives us

$$V_n(1) \approx \frac{\pi^{n/2}}{\sqrt{2\pi(n/2)}\left(\frac{n/2}{e}\right)^{n/2}} = \frac{1}{\sqrt{n\pi}}\left(\frac{2\pi e}{n}\right)^{\frac{n}{2}}$$

For $V_n(r)$ to have a volume of $O(1)$, the radius $r$ must be approximately $\sqrt{n/(2\pi e)}$ asymptotically. Bringing $r = \sqrt{n/(2\pi e)}$ to equation 13, we get

$$\frac{d \cdot V_n(\sqrt{\frac{n}{2\pi e}})}{\exp(\sqrt{\frac{n}{2\pi e}})} \leq Z_t \leq d \cdot V_n(\sqrt{\frac{n}{2\pi e}}). \tag{14}$$

According to equation 2, $N \approx \frac{Ald_{\text{emb}}}{T_{\max}}n$ and $D \approx T_{\max}d$. Therefore, for $Z_t$ to reach $Z_t = 1$, we need $N = O(D^2)$ for well-separated patterns $\mathcal{D}$. The following proposition summarizes the result.

**Proposition 5.** *During memorization of well-separated patterns learned from the data, to minimize the cross-entropy loss, the optimal balance between model size $N$ and data size $D$ is $N = O(D^2)$.*

In Table 2 in Appendix B, we compare the reported cross-entropy loss of various transformer-based models in the literature. Usually, a family of models ranging in a variety of sizes is reported, and we select the largest ones. We observe that similar cross-entropy loss is achieved across a wide range of architectural shapes (including depth, width, attention heads, FF dimensions, and context lengths). Nevertheless, the pre-training cross-entropy losses all satisfy $L > 1$.

**Remark 2.** *We remark that some models add auxiliary regularization terms such as the z-loss (Chowdhery et al., 2023; Yang et al., 2023) during their training. In these cases, the scaling laws should take into consideration the additional terms. Also, modifications to the transformer blocks, such as additional layer normalization may contribute to the lower bound of the cross-entropy.*

### 5.3 SUMMARY OF EXPERIMENTATION

In Appendix F, we perform a series of experiments to validate our theoretical assumptions and results. Below is a concise summary of these experiments.

**Evaluation of the radius with GPT-2** In Appendix F.1, we evaluate the radius $r$ in $Z_l$ of a 24-layer pre-trained GPT-2 *medium* model. Our aim is to validate the hypothesis regarding the radius of patterns in Section 5. We randomly sample 100K chunks, each containing 256 tokens, from the OpenWebText dataset and record the activation vectors from the $l$-th layer. The distance between each activation vector and its nearest neighbor is computed using the Euclidean norm.

**Significance:** The experiment found that the distances between activation vectors in the latent space are approximately equal to the hypothesized magnitude of $2\sqrt{n/2\pi e}$ in equation 14.

**Training vanilla Transformers** In Appendix F.2, we train vanilla Transformers on the Question-Formation dataset, comprising 2M tokens with pairs of English sentences in declarative and question forms. We train two vanilla Transformers with 6 and 10 layers, respectively, each with an embedding dimension of 512, following the configurations from (Murty et al., 2023).

**Significance:** The Question-Formation dataset not only offers a fixed amount of data for evaluating models of different sizes, but also has a limited vocabulary that ensures the well-separated condition in Assumption 3. The training losses stabilize at a value of approximately 1, aligning with the prediction in Proposition 4.

**Training models with varying widths and dimensions** In Appendix F.3, we train models of varying sizes based on the OpenELM design, adjusting the model dimensions and number of heads to achieve different model sizes while maintaining a fixed number of layers. The models have Transformer parameter counts of approximately 40M, 60M, and 80M. These models are pre-trained from scratch using GPT-2 encoding on the Standardized Project Gutenberg Corpus (SPGC) dataset,

with training data sizes ranging from 167.06M to 333.17M tokens. The optimal combinations of model parameters and dataset size are determined by ensuring that training losses plateau and test losses are minimized.

**Significance:** The ratio of model parameters to the square of the dataset size consistently approaches a constant value, supporting the theoretical results in Proposition 5.

## 6    CONCLUSION

We model transformer-based networks with associative memory and study the cross-entropy loss with respect to model and data sizes. We employ a distance-based layer-wise energy function that corresponds to a nearest neighbor search across patterns memorized during training. We then construct a global energy function for the layered structure of the transformer models using the majorization-minimization technique. In practice, we have observed that the majority of transformer models at the commercial level tend to achieve a cross-entropy loss of approximately 2.2. The optimal balance between model and data sizes, however, is often determined by the collective expertise of practitioners. Additionally, the performance of these models can be compromised by both early and delayed stopping. We believe the current paper represents an important step towards understanding the pre-training behaviors of large transformer models. Empirical evidence supporting our study can be found in Appendix F. However, given the significant allocation of computational resources to other scaling law investigations, we acknowledge that our numerical experiments constitute a preliminary evaluation, constrained by computational restrictions. However, it is imperative to underscore the theoretical significance of these findings.

## ETHICS AND REPRODUCIBILITY STATEMENT

**Limitations.**    The theoretical results rely on the assumptions, including Assumption 1 and Assumption 3, made in the main content. We have shown that there exists a dependency between the best model size and the dataset used for training transformer-based models, both in theory and in practice. The most notable limitation is that achieving optimal performance through memorization requires high-quality data. Since most models are trained on data derived from the Internet, the resulting patterns may not be well separated. Future work will be required to identify the relationship between the dataset and the learned patterns. The experiments are conducted on GPT-2, OpenELM models, and vanilla Transformers. We expect that these results generalize to other transformer models. Also, we do not consider other modifications such as lateral connections, skip-layer connections, mixture of experts, mixture of depths, routing, or other compressive modules in the current work. These will be interesting future directions.

**Broader impacts.**    This study elucidates the performance of transformer-based models, measured by cross-entropy loss. Consequently, the findings presented herein have the potential to influence strategic budget allocation and model lifecycle management. By offering insights into the balance between model performance and resource efficiency, it provides insights into the theoretically optimal cross-entropy loss, which can inform both budgetary planning and model termination strategies. We believe that this research delineates a constructive pathway for organizations to foster a more sustainable approach to optimize their machine learning initiatives.

Given that the patterns recognized are directly extracted from the dataset at hand, there exists a potential negative societal consequence: the propensity for models to memorize and thereby perpetuate biases against certain groups. This risk can be attenuated through the implementation of rigorous class-balancing methodologies. Nonetheless, it is imperative that comprehensive fairness assessments be conducted prior to the deployment of any model.

**Reproducibility.**    While the present study is predominantly theoretical in its nature, we have taken multiple steps to ensure the reproducibility of our experiments. We refer the reader to Appendix F and Appendix E for a complete description of the experiments. We have also attached necessary code in the supplementary material.

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

## A    Discussion

**Relationship to the Chinchilla Scaling Laws.**    Our findings are contingent upon the transformer layers' memorization of patterns, which are presumed to be well-separated points learned from the data. In our experiments, we utilize a reduced dataset to emulate the conditions of pattern separation and memorization. Moreover, we have trained the models on their respective datasets repeatedly, ensuring that the training losses have stabilized and the test losses have begun to ascend, indicative of a mild degree of over-parameterization. These conditions diverge from those of the Chinchilla experiment. In practical scenarios, commercial LLMs, akin to the Chinchilla model, are not subjected to such conditions. As we have noted in our paper, it has been observed that even after training on up to 2T tokens, some models have yet to exhibit signs of saturation. In practice, we have observed that the majority of transformer models at the commercial level tend to achieve a cross-entropy loss of approximately 2.2. The optimal balance between model and data sizes, however, is often determined by the collective expertise of practitioners. Additionally, the performance of these models can be compromised by both early and delayed stopping. Therefore, our current experimental setup represents an idealized condition that has not been encountered in commercial LLMs. Nevertheless, considering the substantial computational resources allocated to other scaling law investigations, we recognize that our numerical experiments represent a preliminary assessment that is contingent upon computational limitations, with a thorough analysis reserved for subsequent research endeavors.

**Relationship to Prior Work on Hopfield Networks**    Recently, there has been a growing interest in physics-informed neural networks. The Energy Transformer (Hoover et al., 2024) designs a energy attention mechanism and the corresponding energy function, resulting in a unique model with a strong theoretical foundation that achieves SoTA results on graph anomaly detection and graph classification tasks. Wu et al. (2024) introduces a kernelized version of modern Hopfield networks, aiming to express energy in a feature space where patterns are well-separated, thus avoiding memory interference. Hu et al. (2024b) presents a theoretical framework for deriving and analyzing a family of modern Hopfield models, and Hu et al. (2024a) offers a compression method for Hopfield models, showing superior post-quantization performance compared to vanilla Transformers. While the existing literature, such as Hierarchical Associative Memory Krotov (2021), employs a system of differential equations to design a global energy function that can encompass feedback connections, our proposed model tackles the global energy specific to feedforward architectures. Predictive coding networks (Tang et al., 2023; Li et al.) incorporate recurrent connections; however, their dynamics are focused on minimizing the total squared prediction errors and has less connection to the attention mechanism. By conceptualizing the information retrieval properties of the Transformer as a sequence of associative memories, our model endeavors to establish relationships between model size and memorization (of training data) within the framework of statistical physics.

## Summary of Notations

| | |
|---|---|
| $\rho_i$ | The $i$-th pattern |
| $A$ | Constant depends on the number of layers and the hidden dimension of the network |
| $B_i$ | Ball associated to the $i$-th pattern |
| $d'$ | Number of samples in the test samples |
| $d(\cdot, \cdot)$ | A distance metric |
| $D$ | Size of the training data, $D \approx T_{\max} d$ |
| $d$ | Number of samples in the training samples |
| $d_{\mathrm{emb}}$ | Embedding dimension |
| $L$ | Cross-entropy loss |
| $l$ | Number of Transformer layers |
| $N$ | Number of Transformer parameters |
| $n$ | Dimension of input sequences, $n = T_{\max} d_{\mathrm{emb}}$ |
| $R$ | Radius of the sphere $S$ of patterns |
| $T_{\max}$ | Max number of tokens in a sequence |

## B DEFERRED TABLES

Table 1: Table of selected related works for Hopfield network, enumerating their domain, energy function, and memory capacity. For all the works above, $n$ represents the dimension of the input vector. $W$ is the outer product of the patterns. $M$ is the matrix of patterns. $r$ is the order of polynomial $F(\cdot)$, $d$ is the number of patterns, and $c$ is a positive constant.

| Reference | Domain | Energy | Capacity |
|---|---|---|---|
| Hopfield (1982) | $\{-1, +1\}^n$ | $E(x) = -\frac{1}{2}x^T W x - b^\mathsf{T} x$ | $O(n)$ |
| Krotov & Hopfield (2016) | $\{-1, +1\}^n$ | $E(x) = -\sum_{i=1}^n F((\rho^i)^\mathsf{T} x)$ | $\Theta(n^r)$ |
| Demircigil et al. (2017) | $\{-1, +1\}^n$ | $E(x) = -\text{LogSumExp}(M^\mathsf{T} x)$ | $\Theta(2^{\frac{n}{2}})$ |
| Ramsauer et al. (2020) | $\mathbb{R}^n$ | $E(x) = -\text{LogSumExp}(\beta, M^\mathsf{T} x) + \frac{1}{2}x^T x + \beta^{-1}\log d + \max_i \|\rho^i\|^2/2$ | $\Theta(c^{\frac{n-1}{4}})$ |

Table 2: Transformer-based language models and their reported cross-entropy loss.

| Model | Model Size | Data Size | $L$ | Reference |
|---|---|---|---|---|
| Transformer | 1.5B | 22B | 2.5 | Kaplan et al. (2020) |
| Chinchilla | 70B | 1.4T | 2.2 | Hoffmann et al. (2022a) |
| PaLM 2 | 16B | 100B | 2.4 | Anil et al. (2023) |
| GPT-2 | 8.7B | 178B | 2.3 | Muennighoff et al. (2024) |
| MiniCPM | 2.4B | 140B | 2.4 | Hu et al. (2024c) |
| Nanotron | 1.2B | 105B | 2.4 | Peng et al. (2024) |

Table 3: Mean squared error over 1000 iterations between training loss and minimal validation loss for different model configurations and pre-training settings. The last column reports the ratio between $N$ and $D^2$ for $D^*$ with unit $10^{-10}$.

| MSE | $D = 167.06$M | $D = 190.38$M | $D^* = 214.18$M | $D = 237.98$M | $N/D^{*2}\ (10^{-10})$ |
|---|---|---|---|---|---|
| $N = 39.95$M | 0.07 | 0.05 | **0.04** | 0.04 | 8.71 |

| | $D = 214.18$M | $D = 237.98$M | $D^* = 261.78$M | $D = 285.57$M | $N/D^{*2}\ (10^{-10})$ |
|---|---|---|---|---|---|
| $N = 60.26$M | 0.05 | 0.04 | **0.02** | 0.02 | 8.79 |

| | $D = 261.78$M | $D = 285.57$M | $D^* = 309.37$M | $D = 333.17$M | $N/D^{*2}\ (10^{-10})$ |
|---|---|---|---|---|---|
| $N = 80.20$M | 0.05 | 0.04 | **0.02** | 0.02 | 8.38 |

## C   Some properties of the energy functions

We introduce some useful properties of the LogSumExp function defined below. This is particularly useful because The softmax function, widely utilized in the Transformer models, is the gradient of the LogSumExp function. As shown in (Grathwohl et al., 2019), the LogSumExp corresponds to the energy function of the a classifier.

$$\text{LogSumExp}(x) := \log \sum_{i=1}^{n} e^{x_i}, \quad x = (x_1, \ldots, x_n) \in \mathbb{R}^n.$$

**Lemma 1.** $\text{LogSumExp}(x)$ *is convex.*

*Proof.*

$$t\text{LogSumExp}(x) + (1-t)\text{LogSumExp}(y) = \log\left(\sum_{i=1}^{n} e^{x_i}\right)^t \left(\sum_{i=1}^{n} e^{y_i}\right)^{1-t}$$

$$\geq \log \sum_{i=1}^{n} e^{tx_i + (1-t)y_i} = \text{LogSumExp}(tx + (1-t)y) \quad \forall t \in [0, 1].$$

$\square$

**Lemma 2.** *Suppose* $x = (x_1, \ldots, x_n) \in \mathbb{R}^n$, *then we have*

$$\max_{1 \leq i \leq n} x_i < \text{LogSumExp}(x) \leq \max_{1 \leq i \leq n} x_i + \log n.$$

*Proof.* Taking log on each side of the inequality

$$\exp\left(\max_{1 \leq i \leq n} x_i\right) < \sum_{i=1}^{n} \exp(x_i) \leq \sum_{i=1}^{n} \exp\left(\max_{1 \leq i \leq n} x_i\right)$$

yields the results. $\square$

Consequently, we have the following smooth approximation for the min function.

**Lemma 3.** *Suppose* $x = (x_1, \ldots, x_n) \in \mathbb{R}^n$, *then we have*

$$\min_{1 \leq i \leq n} x_i - \log n \leq -\text{LogSumExp}(-x) < \min_{1 \leq i \leq n} x_i.$$

**Lemma 4.** *For* $x = (x_1, \ldots, x_n), y = (y_1, \ldots, y_n) \in \mathbb{R}^n$, *we have*

$$|\text{LogSumExp}(x) - \text{LogSumExp}(y)| \leq \|x - y\|_{\infty},$$

*where* $\|x\|_{\infty} := \max_{1 \leq i \leq n} |x_i|$.

*Proof.* Let

$$f(t) := \text{LogSumExp}(tx + (1-t)y), \quad \forall t \in [0, 1].$$

According to the mean value theorem, $\exists s \in (0, 1)$ such that

$$\text{LogSumExp}(x) - \text{LogSumExp}(y) = f'(s) = \frac{\sum_{i=1}^{n} \exp(sx_i + (1-s)y_i)(x_i - y_i)}{\sum_{i=1}^{n} \exp(sx_i + (1-s)y_i)}.$$

So

$$|\text{LogSumExp}(x) - \text{LogSumExp}(y)| \leq \frac{\sum_{i=1}^{n} \exp(sx_i + (1-s)y_i)\|x - y\|_{\infty}}{\sum_{i=1}^{n} \exp(sx_i + (1-s)y_i)} = \|x - y\|_{\infty}.$$

$\square$

## C.1 PROOF OF PROPOSITION 2

*Proof.* Let $\xi = \max_{1 \leq i \leq d} \|\rho^i\|$, then we have

$$2E_{\text{MCHN}}^{\beta=2}(x) = -\log\left(\sum_{i=1}^{d} \exp(2(\rho^i)^\intercal x)\right) + \log d + \|x\|^2 + \xi^2$$

$$= -\log\left(\sum_{i=1}^{d} \exp(2(\rho^i)^\intercal x)\right) - \log(\exp(-(\|x\|^2 + \xi^2))) + \log d.$$

So

$$2E_{\text{MCHN}}^{\beta=2}(x) - \log d = -\log(\sum_{i=1}^{d} \exp(2(\rho^i)^\intercal x - \xi^2 - \|x\|^2))$$

$$= -\log(\sum_{i=1}^{d} \exp(\|\rho^i\|^2 - \xi^2 - \|\rho^i - x\|^2)).$$

Therefore, due to Lemma 4, we have

$$|E(x) - (2E_{\text{MCHN}}^{\beta=2}(x) - \log d)| = |\text{LogSumExp}(\|\rho^i\|^2 - \xi^2 - \|\rho^i - x\|^2) - \text{LogSumExp}(-\|x - \rho^i\|^2)|$$

$$\leq \max_{1 \leq i \leq d} |\|\rho^i\|^2 - \xi^2| = \max_{1 \leq i \leq d} \|\rho^i\|^2 - \min_{1 \leq i \leq d} \|\rho^i\|^2.$$

□

# D DEFERRED PROOFS FROM SECTION 5

## D.1 PROOF OF PROPOSITION 4

*Proof.*

$$L(N, D) = H(p_{\widetilde{\mathcal{D}}}, p_\theta) = -\frac{1}{d'} \sum_{x \in \widetilde{\mathcal{D}}} \log(p_\theta(x)) = -\mathbb{E}_{x \sim p_{\widetilde{\mathcal{D}}}}[\log p_\theta(x)]$$

$$= \log Z_\theta \int_{x \in \Omega} P_{\widetilde{\mathcal{D}}}(x) \, \mathrm{d}\mu + \frac{1}{d'} \int_{x \in \Omega} \sum_{i=1}^{d'} \delta(x - \rho^{\sigma(i)}) E_{\text{global}}(x) \, \mathrm{d}\mu$$

$$= \log Z_\theta + \frac{1}{d'} \sum_{\rho^{\sigma(i)}} E_{\text{global}}(x)$$

$$\stackrel{(a)}{=} \log Z_\theta + \frac{1}{Z_t} - \log l + c \stackrel{(b)}{\approx} \log Z_t + \frac{1}{Z_t} \qquad (15)$$

where $(a)$ is because $g(\rho^{\sigma(i)}) = 0$, and $(b)$ is due to equation 10, where we have

$$Z_\theta = \int_{x \in \Omega} \exp(-E_{\text{global}}(x)) \mathrm{d}x = \frac{l}{e^c} \int_{x \in \Omega} \exp(-E_t(x)) \mathrm{d}x, \quad \text{and}$$

$$\log Z_\theta \approx \log l - c + \log \int \exp(-E_t(x)) \mathrm{d}x = \log l - c + \log Z_t.$$

□

## D.2

**Lemma 5.** *The incomplete gamma function $\gamma(n, r)$ satisfies*

$$e^{-r} \frac{r^n}{n} \leq \gamma(n, r) \leq \frac{r^n}{n}$$

*Proof.* For $0 \le x \le r$, we have

$$x^{n-1}e^{-r} \le x^{n-1}e^{-x} \le x^{n-1}.$$

Integrating from 0 to $r$ on each side yields the result. $\qquad\square$

**Theorem D.1** (Stirling's approximation). *For any complex $z$, the Stirling's approximation gives that*

$$\Gamma(z) = \sqrt{\frac{2\pi}{z}} \left(\frac{z}{e}\right)^z \left(1 + O(\frac{1}{z})\right).$$

For large $z$,

$$\Gamma(z+1) \approx \sqrt{2\pi z} \left(\frac{z}{e}\right)^z.$$

## E    TRANSFORMER DETAILS: USING GPT-2 AS AN EXAMPLE

The original GPT-2 model was trained on a 40GB large dataset called WebText that is made of data derived from outbound links from Reddit. The model is trained on the next sentence prediction (NSP) task in a self-supervised manner. A pre-trained tokenizer can be applied to convert the text into tokens using a fixed vocabulary. A max token length $T_{\max}$ (e.g., $T_{\max} = 1024$) is set, so during training, if the number of tokens is greater than $T_{\max}$, the documents will be truncated. The model is trained causally, which means that the prediction for the next token only depends on the inputs from earlier tokens. The model was trained with a global batch size of 512, and the test perplexity still improves if given more training time.

GPT-2 uses a byte-level version of Byte Pair Encoding (BPE), and the vocabulary size is $n_{\text{voc}} = 50,257$. The hidden dimension for the medium size model is $d_{\text{emb}} = 1024$. So the input sequence is of $T_{\max}d_{\text{emb}}$ dimension. These sequences are passed through the model. For the medium size model, these include 24 transformer encoder blocks with 1024 hidden units and 16 self-attention heads (i.e., $l = 24, d_{\text{emb}} = 1024, n_h = 16$). The number of parameters used for word embedding is $n_{\text{voc}} \cdot d_{\text{emb}}$. The number of parameters in the multi-head attention layer is $l \cdot n_h \cdot (3 \cdot d_{\text{emb}} \cdot d_{\text{emb}}/n_h) = 3ld_{\text{emb}}^2 = 75,497,472$. The number of parameters in the dense weights and layer normalization is $l(d_{\text{emb}}^2 + 2d_{\text{emb}}) = ld_{\text{emb}}^2 + 2ld_{\text{emb}} = 25,214,976$, and the number of parameters in the feed-forward weight matrices and bias is $l(2d_{\text{emb}} \cdot d_{\text{FF}} + d_{\text{emb}} + d_{\text{FF}}) = 6ld_{\text{emb}}^2 + 4ld_{\text{emb}} = 151,093,248$, with $d_{\text{FF}} = 3072 = 3d_{\text{emb}}$. As we can observe, the multi-head attention and feed-forward layers account for most of the parameters in the model, and $N \approx Ald_{\text{emb}}^2$ with some constant $A \approx 10$ in this case.

The loss used for the GPT-2 model is the log-probability of a dataset divided by the number of canonical units (e.g., a character, a byte, a word), which is equivalent to the cross-entropy loss. The cross-entropy loss is commonly used to measure the divergence between the predicted probabilities and the true labels. For the NSP task, the model is trained to predict the next token in a sequence based on the context of the previous tokens. So the cross-entropy is taken between the predicted probabilities $\Pr_\theta(x_i)$ of the token $x_i$ and the labels' probabilities $\Pr_D(x_i)$ for all tokens $x_i$ in the vocabulary, i.e.,

$$-\frac{1}{D}\sum_{i=1}^{D}\log(p_\theta(x_i)) = -\sum_{x \sim p_D} p_D(x)\log(p_\theta(x)).$$

Another commonly used loss is the perplexity, which is equivalent to the exponentiated version of the cross-entropy.

## F    EMPIRICAL RESULTS

We explore the hypothesis regarding the radius $r$ in Section 5 using a pre-trained GPT-2 *medium* model. Additionally, we train vanilla Transformers and OpenELM models of different sizes to explore their cross-entropy losses.

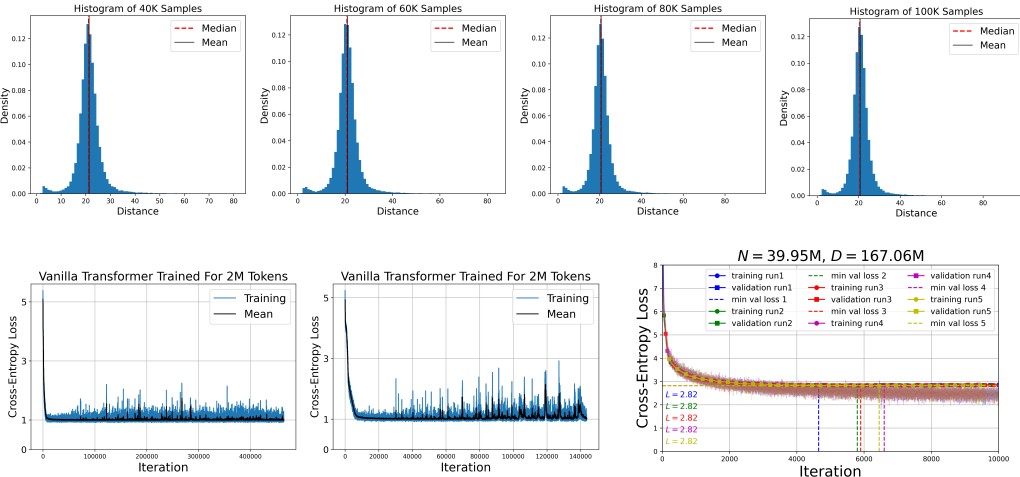

Figure 2: **Top:** Distribution of nearest neighbor distances for output activations utilizing $40\%, 60\%, 80\%$, and $100\%$ of output data. The mean and median values of these distances consistently hover around 20, aligning with the magnitude $2\sqrt{n/2\pi e}$ as hypothesized. **Bottom-left:** Performance of vanilla Transformers with 6 layers (*left*) and 10 layers (*middle*), each trained on the 2M Question-Formation dataset. The models were configured according to the experimental setup detailed in (Murty et al., 2023). The training losses for both models converge to a value of approximately 1, a finding that is consistent with Proposition 4. **Bottom-right:** The pre-training loss (dots) and validation loss (squares) of an OpenELM model across five training runs. The minimal validation losses are displayed in dashed lines. Each run's performance is marked by distinct colors, with the minimum validation loss value for each run indicated along the y-axis.

## F.1 EMPIRICAL EVALUATION OF THE RADIUS

We evaluate the radius $r$ in $Z_l$ of a pre-trained GPT-2 *medium* model. We use the 24-layer pre-trained GPT-2 model (Radford et al., 2019)[1]. The medium size model has 355M parameters. The model is pre-trained with the next sentence prediction task on a large (40 GB) text corpus extracted from web pages. The hidden dimension is $d_{\mathrm{emb}} = 1024$.

We test the model on the OpenWebText (Gokaslan & Cohen, 2019) dataset, a reproduction of the WebText dataset used for training the GPT-2 model. The dataset contains 9B tokens from 8,013,769 documents. We randomly sample 100K chunks of 256 tokens from the dataset. These cover approximately $1.3\%$ of the documents and constitute approximately $0.25\%$ of the tokens used for training. For each sample chunk, we record the activation vector of the last layer for prediction of the next token. As discussed above, each vector should be close to a stored pattern $\rho^{l_i}$. We calculate the distance between each activation vector and its nearest neighbor in terms of the $L_2$ norm and find the nearest neighbor distance for each vector, which results in 100K distance values. In Fig. 2 (top), we plot the histogram of the nearest neighbor distances for these output activations using $40\%, 60\%, 80\%$, and $100\%$ of the output vectors. In all these cases, the mean and median equals approximately to 20, so that a typical $B_i$ of pattern $\rho^i$ has radius 10. This corroborates equation 14 in Section 5, according to which the radius is of order $\sqrt{1024/(2\pi e)} = 7.74$. The activation is only collected for $1.3\%$ of the documents, so the estimated radius may be greater than the actual value.

**Significance:** The experiment's objective is to determine the radius $r$ in $Z_l$ of a pre-trained GPT-2 model, thereby validating the hypothesis regarding the radius of patterns as stated in equation 14. We compute the Euclidean distances between random sequences processed by the GPT-2 *medium* model to estimate the magnitude of the radius $r$. The results indicate that the mean and median of the nearest neighbor distances decrease as the sample size increases, and are around 20, which suggests that a typical $B_i$ associated with the pattern $\rho^i$ has a radius of approximately 10. Considering the relatively small sample size, we anticipate that the actual radius may be smaller than 10. Nevertheless, the

---

[1]available at https://github.com/openai/gpt-2

experiment offers valuable insights into the order of magnitude of the radius and confirms the validity of the hypothesis concerning the radius of patterns within the model's latent space.

## F.2 TRAINING VANILLA TRANSFORMERS

We next train vanilla Transformers using a small amount of high-quality data. The Question-Formation dataset, proposed by McCoy et al. (2020), consists of pairs of English sentences in declarative formation and their corresponding question formation. The dataset contains $D = 2\text{M}$ tokens. The sentences are context-free with a vocabulary size of 68 words, and the task is to convert declarative sentences into questions.

We follow the settings in (Murty et al., 2023) to train two vanilla Transformers ($d_{\text{emb}} = 512, T_{\max} = 5000$) with $l = 6$ layers and $l = 10$ layers respectively. The training losses are shown in Fig. 2 (bottom-left), where the losses stabilize at a value of around $L = 1$ as predicted in Proposition 4.

**Significance:** The experiment investigates the pre-training loss by training two vanilla Transformers with varying depths. Utilizing the Question-Formation dataset, this experiment offers a fixed amount of data to evaluate the model's memorization ability. The dataset's simplicity, featuring a limited vocabulary and a context-free structure, facilitates the well-separated condition in Assumption 3. The stabilization of training losses around a value of 1 not only aligns with the theoretical predictions in Proposition 4 but also indicates that the models have reached a point of diminishing returns, where memorization is likely to predominate.

## F.3 TRAINING MODELS WITH VARYING WIDTHS AND DIMENSIONS

Next, we train models of different sizes following the OpenELM (Mehta et al., 2024) design[2], varying the model configurations including the dimensions and the numbers of heads to achieve different model sizes while keeping the number of layers fixed. We choose different widths and dimensions such that the number of parameters of the transformer layers are about $N =$ 40M, 60M, and 80M respectively. The configurations and hyperparameters can be found in Appendix G. Specifically, the number of layers is fixed in our experiments, as empirical evidence has demonstrated that the depth is a determinant factor influencing the performance.

We utilize the Standardized Project Gutenberg Corpus (SPGC) dataset (Gerlach & Font-Clos, 2020), which contains a filtered timeseries of word-tokens without punctuation, derived from the Project Gutenberg digital library of public domain literary works. We choose this subset because it offers a collection of high-quality word sequences. We prepare the training data using different proportions of the first 180M words of the SPGC, and we use the last 5% tokens as the validation set. We pre-train the models from scratch on the tokenized training data with GPT-2 encoding, which encompasses eight distinct sizes ranging uniformly from $D =$ 167.06M to $D =$ 333.17M. Throughout the training, we employ random sampling to select chunks of the pre-determined context length. Given the typically extensive length of the e-books within this dataset, it is plausible that sequences drawn with the context length of 1024 tokens originate from the same book.

We report the number of transformer parameters $N$ and the corresponding $D^*$ such that training loss plateaus and the test loss is minimized, defined by $D^* = \min\{D : \text{MSE}(L_{\text{train}}^{(N,D)}, L_{\min}^{(N,D)}) < \sigma^2\}$, where $L_{\text{train}}^{(N,D)}$ is the training loss of model of size $N$ using training data of size $D$, $L_{\min}^{(N,D)}$ is the minimal validation loss throughout the training steps (as is depicted in the bottom-right of Fig. 2), and MSE is the mean squared error taken over the proceeding 1000 iterations of the step where the validation loss is minimized. In Table 3, we report the MSE for each configuration, with $D^*$ highlighted in each row when $\sigma^2 = 0.04$. Upon analyzing the optimal combinations, we have computed the ratio of model parameters to the square of the dataset size, as demonstrated in the last column. The threshold $\sigma$ were empirically selected based on our preliminary experiments, as provided in Appendix G for visual inspection. Our findings indicate that the ratio $N/D^{*2}$ consistently approaches a constant value, reinforcing our theoretical results in Section 5.

**Significance:** The experiment aims to support Proposition 5, which predicts a quadratic relationship between the number of Transformer parameters and the size of well-separated datasets. We train

---

[2]available at https://github.com/apple/corenet.

models of varying widths and dimensions following the OpenELM design from scratch and adjust the training data size between 167.06M and 333.17M. We identify the optimal combinations of model parameters and dataset size by ensuring that the training losses plateau and the test losses are minimized. The consistent approach of the ratio of model parameters to the square of the dataset size towards a constant value corroborates Proposition 5, implying that an optimal ratio exists, which can inform the choice of model size in relation to dataset size for optimal performance.

# G    EXPERIMENTAL DETAILS

## G.1    CONFIGURATIONS

We follow the OpenELM (Mehta et al., 2024) architecture and choose the following configurations such that the number of transformer parameters are about 40M, 60M, and 80M.

| Parameter | Model 1 | Model 2 | Model 3 |
|---|---|---|---|
| Number of Transformer Parameters | 39.95M | 60.26M | 80.20M |
| Model Dimension | 954 | 1280 | 1440 |
| Number of Transformer Layers | 8 | 8 | 8 |
| Number of KV Heads | 3, 3, 3, 3, 4, 4, 4, 5 | 3, 3, 3, 3, 4, 4, 4, 5 | 3, 3, 3, 4, 4, 4, 5, 5 |
| Number of Query Heads | 6, 6, 6, 6, 8, 8, 8, 10 | 6, 6, 6, 6, 8, 8, 8, 10 | 6, 6, 6, 8, 8, 8, 10, 10 |

Hyperparameters for pre-training are listed below.

| Parameter | Detail |
|---|---|
| Tokens per iteration | 491,520 |
| Vocabulary Size | 50,257 |
| Activation Function | swish |
| Attention Dropout | 0.1 |
| Embedding Dropout | 0.1 |
| Head Dimension | 64 |
| Initializer Range | 0.02 |
| Max Context Length | 1024 |
| Normalization Layer | rms_norm |
| Normalize QK Projections | True |
| QKV Multipliers | 0.5, 1.0 |
| Batch size | 12 |
| AdamW | $\beta_1 = 0.9, \beta_2 = 0.95, \epsilon = 10^{-8}$ |

## G.2    ROBUSTNESS TO DEDUPLICATION

In order to further confirm the validity of our analyses, we conduct five independent training runs of Model 1 to assess the model's robustness and the consistency of its performance across different training instances. Fig. 3 illustrates the CE loss over the course of training iterations for each run, along with the minimum validation loss achieved during each run. It can be observed that the training runs exhibit varying degrees of performance, as indicated by the different trajectories of the CE loss curves. The minimum validation loss remains consistent to the second decimal place across various training runs, and is achieved at approximately the 6000th training step.

## G.3    ADDITIONAL RESULTS

Figured below are the training dynamics of the models in Table 3 for visual inspection.

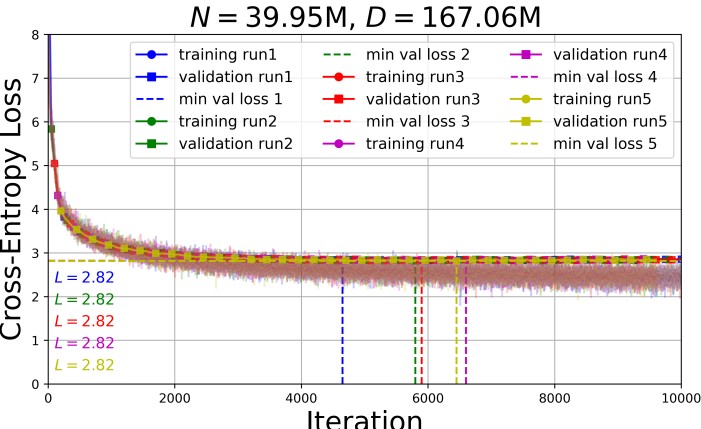

Figure 3: The cross-entropy loss for one model configuration during pre-training (depicted with dots) and validation (depicted with squares) across five separate training runs. The minimal attainable validation loss is represented by dashed lines. Each individual run's performance is distinguished by a unique color, and the y-axis highlights the lowest validation loss for each respective run.

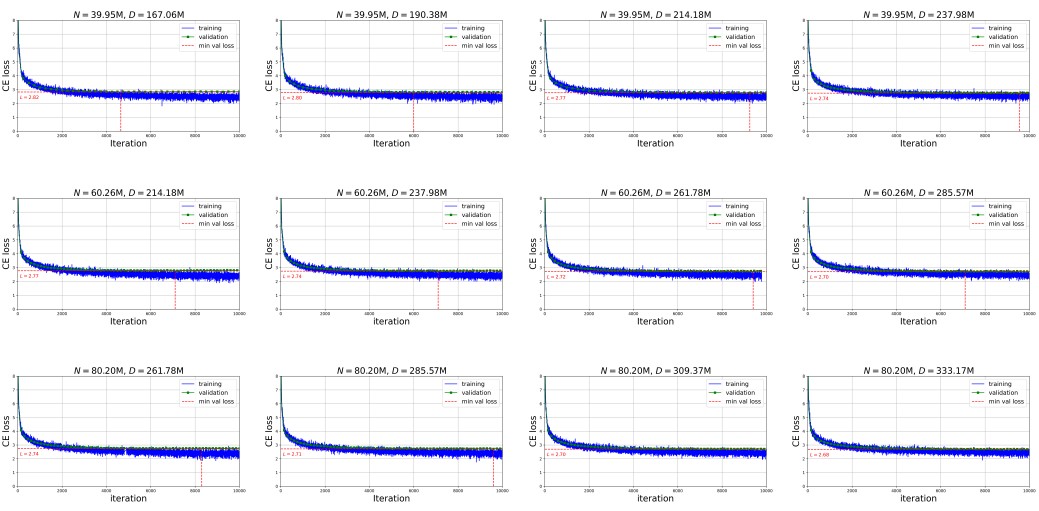

Figure 4: Cross-entropy losses of eight models employing the OpenELM architecture as presented in Table 3.

