# OpenReview forum: "Beyond Scaling Laws: Understanding Transformer Performance with Associative Memory"
_ICLR.cc/2025/Conference — Submitted to ICLR 2025_

### Official Review · Reviewer_JTCX · 2024-10-22

**Soundness:** 4
**Presentation:** 3
**Contribution:** 3
**Rating:** 8
**Confidence:** 4

**Summary:**

This paper looks at the performance of transformer-architecture models, particularly looking at the performance of such models with respect to the number of parameters. Recently proposed scaling laws may not reflect the true effect of parameter count on performance, and some smaller transformers outperform larger ones. The authors employ associative memories to model the behavior of transformers, specifically modelling each transformer unit with a single Hopfield network. Individual Hopfield energies are analyzed in a fashion similar to the Majorization Minimization technique. This allows an energy function to be associated with a transformer block, from which the paper forms a "global energy function" over the entire model. The global energy function relates the transformer size, the dataset size, and the optimal performance of the model. In particular, the authors find that the optimal loss is bounded by a constant term. Experiments are conducted that corroborate these results using a variety of transformer architectures on several datasets. Although some of these experiments were bounded by the computational resources (the authors note the work could be considered preliminary investigations) the results are still seemingly in agreement with the theoretical predictions made in previous Sections.

**Strengths:**

This paper offers a theoretical analysis of the optimal loss for transformer-architecture models, which is corroborated in practical experiments. A novel energy function for the Dense Associative Memory (which the paper refers to as the Modern Continuous Hopfield Network) is proposed, which gives interesting behaviors in comparison to the original energy function from Ramsauer et al. 2020. The work is robustly proven with mathematical analysis that is rooted in both statistical mechanics and prior work on associative memories, and is hence of great quality. The clarity of the paper is reasonable, with the caveat that much of the theoretical work is based in mathematics that is reasonably dense, and is often deferred to the appendix. The theoretical findings appear very significant in the context of transformer-architecture models, although the experimental results may not be as strong in this regard.

**Weaknesses:**

Several decisions / assumptions of the paper are made without discussion that would be useful for the reader. For example, it seems that all training set patterns to be stored explicitly in memory for the proposed energy function to be implemented, which is contrary to the traditional understanding of an associative memory. Almost all of the mathematical proofs are relegated to appendices, which is somewhat disappointing considering how vital these are to the main body of research.

**Questions:**

A slight nitpick, but on line 060 the authors state that the Hopfield network has only been recently generalized to continuous values. However, the continuous value Hopfield network was introduced only shortly after the 1982 paper. See Hopfield, J. "Neurons with graded response have collective computational properties like those of two-state neurons", 1984. The Dense Associative Memory (Modern Hopfield Network), introduced by Krotov and Hopfield 2016, was generalized to continuous values by Ramsauer et al. 2020, so perhaps this is what the authors intended? Reading on, this is discussed more in Section 2, subsection on Hopfield models, so my apologies for citing literature the authors are aware of! A slight rephrasing of line 060 may still be prudent.

The discussion on Hopfield models in Section 2 is excellent, but towards the end (e.g. works on the Energy Transformer and U-Hop networks) are seemingly only tangentially related to the main text. These could be left in for completion, but are they necessary for a reader's understanding of the main text? Perhaps with more explicit ties in later sections these would make more sense for a reader.

Section 3 lays out the model architecture. Section 3.1 discusses the associative memory formulation and how sequences of tokens may be interpreted as patterns in associative memories. Since each prediction will have access to a different number of previous tokens (e.g. predicting token 5 will have access to 4 previous tokens, predicting token 6 will have access to the previous 5, and so on) does this mean the associative memory here must be capable of handling retrieval for patterns of different lengths? Or are all patterns to be padded to a uniform length, such as in a traditional transformer, in which was the associative memory would have a much lower proportion of initial contents for subsequences that are early in the task (e.g. predicting the 5th token compared to predicting the 500th)? Perhaps I am misunderstanding the associative memory structure or pattern structure here? The initial paragraph of Section 3 seems to imply the latter, that all patterns are equal to the maximum token length. Is this correct? If so, how does this affect the retrieval of patterns that have only a small number of tokens available?

Definition 1 is a nice formalization of storage and retrieval in (continuous) associative memories. The condition that the intersection of any two balls must be null seems superfluous, however. If the intersection of B_i and B_j exists that would imply that the same point (i.e. any point in the intersection) would converge to both rho^i and rho^j, which is impossible. Unless this formalization of the associative memory somehow allows for multiple points of convergence? Is the condition of null intersection required for this definition, or could it be removed / discusses as a consequence of the requirement of convergence?

In reference to Assumption 2, are we free to assume that the held-out samples are likely to be stored in a similar fashion to the training set? In typical associative memories even samples from the same distribution as the training set are not stored, instead patterns from the training set have their basin of attraction (here, associated ball) expand to cover the hyper-volume of the validation distribution --- does this intuition not carry over to this architecture for some reason? If not, why not?

The main difference between the MCHN energy function and the proposed energy function appears to be the removal of the regularization terms. Instead, the negative distances are fed into the LogSumExp function, rather than the pattern matrix times the probe vector. To my understanding, this implies that the proposed energy function requires all patterns in the training set to be kept explicitly in memory such that the calculation in Equation 3 can be performed. Later, in Equation 4, the energy function near the patterns of the training set is replaced by the nearest neighbor search which again requires knowledge of all training patterns. Is this correct? In typical associative memories we try to avoid storing the training set patterns directly, as this would defeat the purpose of a model that can learn to retrieve items. However, this paper seems to use the nearest-neighbor-search explicitly which is relatively counterintuitive in the context of associative memories as a whole. The formalization of the search as an energy function is interesting, but I am not yet convinced this leads to a useful associative memory --- if we have the training set explicitly stored, why not conduct a nearest-neighbor search over this data instead of using the energy function?

Assumption 3 references back to the condition that the all balls around training set patterns are well separated. Is this not repeated from the discussion in Section 3? The authors also discuss briefly what this assumption means in practice, but is this property common in real datasets? Or is this a property that is nice to reason about, but in practice can be violated somewhat? It feels that the requirement in Assumption 3 limits the proposed energy function to only very nicely behaved data --- is that correct?

In Section 6.1, from line 465 onwards, the experimental radius of each ball is determined to be 10. Equation 12 would predict this radius to be 7.74. This seems like a fairly large discrepancy; some 30% larger than expected. Is this large a discrepancy expected in practice? The final line of the Section offers some respite, but perhaps increasing the number of documents to verify that the presented radius is larger that the actual value would be useful.

---

> ### Author Response · Authors · 2024-11-19
>
> We thank the reviewer for the careful reading of the paper and constructive suggestions and feedback. We provide our response to the reviewer’s individual comments here, and strongly encourage the reviewer to check our global response for updates regarding associative memory formulation, which is also relevant to the reviewer’s comments. Please find our response to the questions below:
>
> > A slight nitpick, but on line 060 the authors state that the Hopfield network has only been recently generalized to continuous values. However, the continuous value Hopfield network was introduced only shortly after the 1982 paper. See Hopfield, J. "Neurons with graded response have collective computational properties like those of two-state neurons", 1984. The Dense Associative Memory (Modern Hopfield Network), introduced by Krotov and Hopfield 2016, was generalized to continuous values by Ramsauer et al. 2020, so perhaps this is what the authors intended? Reading on, this is discussed more in Section 2, subsection on Hopfield models, so my apologies for citing literature the authors are aware of! A slight rephrasing of line 060 may still be prudent.
>
> We appreciate the suggestion. Indeed, we recognize that the original phrasing could be misleading, and it has been corrected in our revised manuscript (lines 60-61). Additionally, we have expanded the "Related Work" section to incorporate more recent literature.
>
> > The discussion on Hopfield models in Section 2 is excellent, but towards the end (e.g. works on the Energy Transformer and U-Hop networks) are seemingly only tangentially related to the main text. These could be left in for completion, but are they necessary for a reader's understanding of the main text? Perhaps with more explicit ties in later sections these would make more sense for a reader.
>
> We appreciate the suggestion. In response, we have relocated certain related work to the appendices, thereby creating more space for the mathematical components of the paper.
>
> > Section 3 lays out the model architecture. Section 3.1 discusses the associative memory formulation and how sequences of tokens may be interpreted as patterns in associative memories. Since each prediction will have access to a different number of previous tokens (e.g. predicting token 5 will have access to 4 previous tokens, predicting token 6 will have access to the previous 5, and so on) does this mean the associative memory here must be capable of handling retrieval for patterns of different lengths? Or are all patterns to be padded to a uniform length, such as in a traditional transformer, in which was the associative memory would have a much lower proportion of initial contents for subsequences that are early in the task (e.g. predicting the 5th token compared to predicting the 500th)? Perhaps I am misunderstanding the associative memory structure or pattern structure here? The initial paragraph of Section 3 seems to imply the latter, that all patterns are equal to the maximum token length. Is this correct? If so, how does this affect the retrieval of patterns that have only a small number of tokens available?
>
> We appreciate the insightful question from the reviewer. It is true that transformer-based models operate within a predefined maximum context length, and during pre-training, input texts are indeed segmented into sequences of this maximum token length. You are correct in your understanding that in Section 3, we intend to pad the sequences to a uniform length.
> In the context of associative memory, when only a small number of tokens are provided, the model updates by aligning the padded input with the nearest stored pattern. This ensures that the associative memory can effectively handle retrieval for patterns of varying lengths in the latent space. Based on your insights, we have refined our assumptions.
> We have expanded upon this discussion in the global response to provide a more detailed explanation of how our model handles patterns in the latent space.

---

> > ### Author Response · Authors · 2024-11-19
> >
> > > Definition 1 is a nice formalization of storage and retrieval in (continuous) associative memories. The condition that the intersection of any two balls must be null seems superfluous, however. If the intersection of B_i and B_j exists that would imply that the same point (i.e. any point in the intersection) would converge to both rho^i and rho^j, which is impossible. Unless this formalization of the associative memory somehow allows for multiple points of convergence? Is the condition of null intersection required for this definition, or could it be removed / discusses as a consequence of the requirement of convergence?
> >
> > We appreciate the reviewer's insightful suggestion to generalize Definition 1 to account for the potential intersection of B_i and B_j. In response, we have considered the approach of introducing a relaxation similar to Equation (7) in (Saha et al., 2023), which involves partial cluster assignments and aligns with the collective attraction concept proposed in that work. To reflect this consideration, we have included Remark 1 in our revision below Definition 1 (lines 186-189). Given that our theoretical results are built upon prior work, we would like to maintain consistency with the definitions of storage and retrieval as presented in (Ramsauer et al., 2020).
> >
> > * Saha, B., Krotov, D., Zaki, M. J., \& Ram, P. (2023). End-to-end differentiable clustering with associative memories. In International Conference on Machine Learning. PMLR.
> >
> >
> > > The main difference between the MCHN energy function and the proposed energy function appears to be the removal of the regularization terms. Instead, the negative distances are fed into the LogSumExp function, rather than the pattern matrix times the probe vector. To my understanding, this implies that the proposed energy function requires all patterns in the training set to be kept explicitly in memory such that the calculation in Equation 3 can be performed. Later, in Equation 4, the energy function near the patterns of the training set is replaced by the nearest neighbor search which again requires knowledge of all training patterns. Is this correct?  In typical associative memories we try to avoid storing the training set patterns directly, as this would defeat the purpose of a model that can learn to retrieve items. However, this paper seems to use the nearest-neighbor-search explicitly which is relatively counterintuitive in the context of associative memories as a whole. The formalization of the search as an energy function is interesting, but I am not yet convinced this leads to a useful associative memory --- if we have the training set explicitly stored, why not conduct a nearest-neighbor search over this data instead of using the energy function?
> >
> > We appreciate the reviewer's insightful question regarding the differences between the MCHN energy function and our proposed energy function, particularly concerning the handling of patterns in associative memories. Firstly, we acknowledge the distinction between traditional associative memories and our approach. Unlike address-based memories, the goal of associative memory is to retrieve the closest point given a query that is approximately the same type as the data it stores. We agree that in typical associative memories, storing the training set patterns directly would defeat the purpose of a model designed to learn and retrieve items. In our paper, we are studying the collective behavior of the attention layers, and thus we propose an energy function to approximate the underlying mechanism. As demonstrated in Propositions 1, 2, and 3, both the proposed energy and the MCHN energy approximate the auxiliary energy function associated with the nearest-neighbor search. This searching mechanism serves as a medium that connects attention and the proposed energy function. Regarding the storage of patterns, we have revised our assumptions to clarify that the patterns are stored in the latent space as representations, using the concept of storing patterns implicitly rather than explicitly. We hope this clarification addresses your concerns. We have expanded on these points in the global rebuttal to further elaborate on this difference.

---

> > > ### Author Response · Authors · 2024-11-19
> > >
> > > > Assumption 3 references back to the condition that the all balls around training set patterns are well separated. Is this not repeated from the discussion in Section 3? The authors also discuss briefly what this assumption means in practice, but is this property common in real datasets? Or is this a property that is nice to reason about, but in practice can be violated somewhat? It feels that the requirement in Assumption 3 limits the proposed energy function to only very nicely behaved data --- is that correct?
> > >
> > > We appreciate the reviewer's question regarding Assumption 3 and its implications for the separation of patterns in real datasets. Your observation is incisive, and we agree that the assumption of well-separated patterns is an idealized condition. In response to your concern, we acknowledge that real-world datasets often exhibit patterns that are not perfectly separated. The condition referenced in Assumption 3 is indeed a simplification that facilitates theoretical analysis and reasoning about the behavior of our proposed energy function. To address your question about the practicality of this assumption, we have revised the presentation of Assumption 3 to emphasize that the embeddings, post-initial layer processing, exhibit reduced correlation, which aligns with the assumption of pattern separability being less stringent in practice.
> > > We hope this revised explanation clarifies the role of Assumption 3 in our analysis.
> > >
> > >
> > > > In Section 6.1, from line 465 onwards, the experimental radius of each ball is determined to be 10. Equation 12 would predict this radius to be 7.74. This seems like a fairly large discrepancy; some 30\% larger than expected. Is this large a discrepancy expected in practice? The final line of the Section offers some respite, but perhaps increasing the number of documents to verify that the presented radius is larger that the actual value would be useful.
> > >
> > > We appreciate the reviewer's suggestion. We have expanded our experimental dataset to include a larger number of sequences. This increase in data size allows for a more robust verification of the radius value.
> > > Due to the constraints imposed by our current computational resources, we have not been able to conduct experiments on a larger scale at the moment.
> > > Please also refer to our global response. We believe that these revisions strengthen the experimental findings. Thank you again for your valuable suggestions.

---

> > > > ### Comment · Reviewer_JTCX · 2024-11-20
> > > > **Response to Authors**
> > > >
> > > > I thank the authors for their comprehensive reply. The addition to the literature review, and shifting of some topics to the appendix, is a welcome change to the manuscript. The increased consistency with prior work is also appreciated. I also value the authors experiments with increased data volumes, and understand that further experiments would be extremely computationally expensive.
> > > >
> > > > On padding the input to a predetermined length, I appreciate that the associative memory will update to align the pattern with the nearest stored memory (at least, most of the time), but my point is that sequences later in time will have more tokens, hence require less padding, and will have more information for the associative memory to infer on. Sequences from very early in time will have a very small amount of information, which could cause problems with the associative memory, as you are effectively embedding the sequence into a much lower dimension subspace (assuming padding with zeros). Is this not a concern, that the amount of information in a pattern is not uniform across sequences?
> > > >
> > > > Due to the immense quantity of alterations that improve this manuscript I am happy to increase my confidence in my review.

---

> > > > > ### Author Response · Authors · 2024-11-20
> > > > >
> > > > > We are grateful for the reviewer's expeditious feedback and recognition of the improvements made to the manuscript.
> > > > >
> > > > > Regarding the concern about the non-uniformity of sequence lengths and its impact on the associative memory, we thank the reviewer for raising this important point. As demonstrated in (Svete and Ryan, 2024), a Transformer can be modeled as a representation-based n-gram model, as shown in Equation 2 of their work, where the conditional probabilities for the next token are derived from attending to the previous n-1 tokens. This aligns with the intuition that attention scores for earlier tokens diminish in a sequence, and n is expected to be relatively small. For instance, in (Svete et al., 2024), n is experimentally set to the values {2, 4, 6, 8, 12}, which are much smaller than the context length. Consequently, only a small number of padded sequences are affected by the information discrepancy due to non-uniform lengths.
> > > > >
> > > > > In response to this discussion, we have updated the manuscript to include these considerations, now reflected in lines 159-165. We believe that this addition clarifies our approach and addresses the reviewer's concern. Thank you again for your insightful observations. We welcome any further questions and are committed to improving the manuscript's clarity.
> > > > >
> > > > >
> > > > > * Svete, A. and Ryan, C. (2024). Transformers Can Represent n-gram Language Models. In Proceedings of the 2024 Conference of the North American Chapter of the Association for Computational Linguistics.
> > > > >
> > > > > * Svete, A., et al. (2024). Can Transformers Learn n-gram Language Models? In Proceedings of the 2024 Conference on Empirical Methods in Natural Language Processing.

---

### Official Review · Reviewer_zNBU · 2024-11-02

**Soundness:** 4
**Presentation:** 4
**Contribution:** 3
**Rating:** 6
**Confidence:** 3

**Summary:**

Nowadays, it is common to scale up a Transformer model to achieve great performance across a variety of tasks. However, increasing the size of the model does not necessarily lead to performance improvement. Meanwhile, an interesting phenomenon occurs--as the model's ability to memorize the training samples so does its generalization ability. Consequently, this work provides a theoretical framework which elucidates the memorization process in the Transformer model via the lens of Modern Associative Memory or Hopfield models. The contributions include a novel analysis of memorization in each Transformer block as a function of modern Hopfield energy, and mathematical illustrations of the dependency of the size of the model and dataset respectively that is needed for the model to attain optimal performance. Such illustrations highlight the optimal cross entropy loss is bounded by a constant.

**Strengths:**

1. The paper highlights the characteristics (i.e., pros and cons) of the Modern Hopfield Network's energy function given high dimensionality and continuous values, while providing an alternative energy function which maintains convexity. This energy function, i.e., logsumexp of a euclidean distance between a target $x$ and a stored pattern $\rho^{(i)}$, describes the dynamic of each Transformer block as an associative memory retrieval system.

2. Using the above (layer-wise) energy function, the paper also introduces a global energy function which accounts for the stacking of homogenous blocks in Transformer model. This particular global energy function enables the description of probability density function of the entire Transformer model. Using this description, the paper works out the optimal condition for the partition function $Z$ relating back to the scaling law from [Hoffmann et al (2022)](https://proceedings.neurips.cc/paper_files/paper/2022/file/c1e2faff6f588870935f114ebe04a3e5-Paper-Conference.pdf).

3. The paper also provides fantastic mathematical descriptions of memorization using their proposed energy function.

**Weaknesses:**

1. The authors might not be aware of this paper due to its title. But the proposed energy function (eq. 3) in the paper was recently introduced/used in [Saha et al. (2023)](https://proceedings.mlr.press/v202/saha23a/saha23a.pdf), with the exception that they use $\beta$ in their formulation and descriptions of the function's properties. However, I do not think this fact compromises the novelty of the work.

2. Much of the current analyses of storage capacity in Hopfield models relies heavily on uncorrelated patterns. Consequently, the claim of exponential capacity does not necessarily hold true for natural (or correlated) data. Similarly, a weakness of this work is assumption 3, since the work focuses on text data. The authors can instead make a claim that the Transformer blocks are storing the latent data, obtained from passing the text data to an embedding layer ---which should be much more uncorrelated--- to reinforce their narrative.

3. The writing for the experimentation is unclear, especially to unfamiliar readers. For example, I think the authors should elaborate more on the findings of the mean and median nearest neighbor distances, and why such results corroborate eq. 12. Finally, I'm not sure what section 6.3 is trying to demonstrate. How was $\sigma$ chosen? To find $D^*$, you would have to train multiple models across a variety of data sizes $D^{(k)}$ for $k=1...M$ sizes. Perhaps, there should be an elaboration on the selection of the size of different models and each chosen data sizes in relation to $N \approx \frac{A l d_{max}}{T_{max}} n$ and $D = T_{max} n$. The inclusion of these details should fortify the reasonings behind these experiments.

**Questions:**

1. Is it possible to collect the activation outputs up to 5% of the documents? If it's not possible due to limited compute, I understand. However, I think collecting activation outputs from 1% of the documents might be too limited.

2. I believe you forgot to include the citation for RMSNorm, see the paragraph which describes the hypersphere volume on page 8 of the manuscript and where you mentioned LayerNorm.

3. In the description for the cross entropy loss $L$ in Proposition 4, you should perhaps replace that equation with eq. 13 in appendix D, since there is a mention of $c$ which is not found in the equation in the main text.

4. There is an inconsistency in labeling the equations in the main text. Please organize them better since the mathematical descriptions provided in this paper are great. For example, I understand that $N \approx \frac{A l d_{max}}{T_{max}} n$ and $D = T_{max} n$ is mentioned in the prior sections of the manuscript. It is still certainly an important equation to be labeled/numbered.

---

> ### Author Response · Authors · 2024-11-19
>
> We appreciate the reviewer's time and constructive feedback. Your insightful suggestions have been invaluable in enhancing our manuscript. We provide our response to the reviewer’s individual comments here, and strongly encourage the reviewer to check our global response for updates which is also relevant to the reviewer’s comments. Below, we provide a detailed response to each of the comments.
>
> > The authors might not be aware of this paper due to its title. But the proposed energy function (eq. 3) in the paper was recently introduced/used in Saha et al. (2023), with the exception that they use $\beta$ in their formulation and descriptions of the function's properties. However, I do not think this fact compromises the novelty of the work.
>
> We appreciate the reviewer's recommendation. Indeed, we were previously unaware of this research. We have now expanded our literature review in the revision. Specifically, we have added a reference (Lines 137-140) to the independent introduction of the layer-wise energy function equation (3) by Saha et al. (2023), which was presented within the context of clustering. Your suggestion also inspired us to discuss the collective attraction mechanism in Remark 1 (Lines 186-189).
>
> > Much of the current analyses of storage capacity in Hopfield models relies heavily on uncorrelated patterns. Consequently, the claim of exponential capacity does not necessarily hold true for natural (or correlated) data. Similarly, a weakness of this work is assumption 3, since the work focuses on text data. The authors can instead make a claim that the Transformer blocks are storing the latent data, obtained from passing the text data to an embedding layer ---which should be much more uncorrelated--- to reinforce their narrative.
>
> We sincerely appreciate the excellent suggestion. In response, we have revised our assumptions to incorporate the recommended claim. Please also refer to "Regarding the assumptions" in our global rebuttal.
>
> > The writing for the experimentation is unclear, especially to unfamiliar readers. For example, I think the authors should elaborate more on the findings of the mean and median nearest neighbor distances, and why such results corroborate eq. 12. Finally, I'm not sure what section 6.3 is trying to demonstrate. How was $\sigma$ chosen? To find $D^*$, you would have to train multiple models across a variety of data sizes $D^{(k)}$ for $k=1,\ldots,M$ sizes. Perhaps, there should be an elaboration on the selection of the size of different models and each chosen data sizes in relation to $N$ and $D=T_{max}n.$ The inclusion of these details should fortify the reasonings behind these experiments.
>
> We appreciate your suggestions. We have elaborated our method in lines 477-479. In Section 6.3, our objective is to investigate the relationship between the number of transformer parameters $N$ and the data size $D$. To find $D^*$, we trained multiple models across a variety of data sizes $D^{(k)}.$ Due to space constraints, the data sizes $D^{(k)}$ and their corresponding results are presented in Table 3 in the appendix. Both the data sizes $D^{(k)}$ and the threshold $\sigma$ were empirically selected based on our preliminary experiments, as depicted in Appendix F.3, where a grid search was conducted over $D$ and $\sigma$.
>
> > Is it possible to collect the activation outputs up to 5\% of the documents? If it's not possible due to limited compute, I understand. However, I think collecting activation outputs from 1\% of the documents might be too limited.
>
> Thank you for the suggestion. We have managed to updated our experimental result using 1.3\% of the documents. Unfortunately, the scale of our experiments is constrained by our computational resources. Please also refer to our responses on empirical results in the global response.
>
> > I believe you forgot to include the citation for RMSNorm, see the paragraph which describes the hypersphere volume on page 8 of the manuscript and where you mentioned LayerNorm.
>
> We appreciate the reviewer's observation. We have added the citation in our revision in line 407.
>
> > In the description for the cross entropy loss $L$ in Proposition 4, you should perhaps replace that equation with eq. 13 in appendix D, since there is a mention of $c$ which is not found in the equation in the main text.
>
> Thank you for the suggestion. The variable $c$ in equation 13 is now mentioned in equation 10 (originally equation 9) in section 5.
>
> > There is an inconsistency in labeling the equations in the main text. Please organize them better since the mathematical descriptions provided in this paper are great. For example, I understand that $N$ and $D$ are mentioned in the prior sections of the manuscript. It is still certainly an important equation to be labeled/numbered.
>
> Thank you for the suggestion. We have numbered these equations (line 152), and we have further provided a summary of notations in Appendix A.

---

> ### Comment · Reviewer_zNBU · 2024-11-21
>
> Thank you to the authors for their detailed response. After reading over the revised manuscript, I am happy to say that I am considering raising my score, ***but there are a couple of more things that should be addressed before I do so***. I understand this might be over the top but for the sake of improving the quality of the manuscript, it is essential for us (authors and reviewers) to do so.
>
> My first qualm, that has not been addressed, is related to section 4 of the manuscript. Firstly, thank you to the authors for mentioning the work [Saha et al. (2023)](https://arxiv.org/pdf/2306.03209) in the revised version. However, it is clear that ***eq. 4*** is not a new energy function by any means, even though it does not include the inverse temperature as in [Saha et al. (2023)](https://arxiv.org/pdf/2306.03209); also, I am not claiming that their work is the first to introduced this energy function. Instead, I believe the authors should rely heavily on ***eq. 7*** as their novel energy function, which is a global energy function that captures the probability density of all transformer blocks. I believe all of the formulations, done in the work, which lead up to ***eq. 7*** is fantastic and it should be the focus of section 4. Similarly, the same can be said with [Millidge et al. (2022)](https://arxiv.org/pdf/2202.04557) as I did with [Saha et al. (2023)](https://arxiv.org/pdf/2306.03209).
>
> Secondly, I somewhat understand the experimentation, but what I do not get is the significance of the results after reading the manuscript. Specifically, why do these results tie up with the theoretical contributions and why are they significant to the claims made in the abstract. I do not think that the authors have made it very clear to me and possibly, to the other reviewers too. ***I suggest that the authors should have a separate discussion section where they discuss the significance of their results***. Also, for context, after reading lines 477-479, they did not give me that sense anyhow.
>
> Thirdly, in the Hopfield section of the related work section, the authors should use **Millidge et al. (2022)** and **Saha et al. (2023)** instead of **Millidge et al.** and **Saha et al.**. The same can be said in **Remark 1**. This is simply a stylistic comment.

---

> > ### Author Response · Authors · 2024-11-23
> >
> > We sincerely appreciate the reviewer's insightful feedback. We are grateful for the opportunity to enhance our manuscript further; we have implemented substantial revisions in response to your recommendations. We would like to invite the reviewer to consult our comment addressed to all reviewers. Below, please find our detailed responses to your comments.
> >
> > > My first qualm, that has not been addressed, is related to section 4 of the manuscript. Firstly, thank you to the authors for mentioning the work Saha et al. (2023) in the revised version. However, it is clear that eq. 4 is not a new energy function by any means, even though it does not include the inverse temperature as in Saha et al. (2023); also, I am not claiming that their work is the first to introduced this energy function. Instead, I believe the authors should rely heavily on eq. 7 as their novel energy function, which is a global energy function that captures the probability density of all transform blocks. I believe all of the formulations, done in the work, which lead up to eq. 7 is fantastic and it should be the focused on section 4. Similarly, the same can be said with Millidge et al. (2022).
> >
> >
> > Thank you for your suggestion on highlighting the novelty of our global energy function. We concur that Eq. 4 does not represent a new energy function. Accordingly, we have removed all references to the novelty of the layer-wise energy function from our paper, including those previously in the abstract, the contribution, the conclusion, Remark 1, Figure 1, Proposition 1, and the specific lines 250, 267, 274, 286, and 304.
> > Additionally, we have updated our discussion on the inverse temperature in Saha et al. (2023) and the relationship to Millidge et al. (2022), which can now be found in lines 281-283.
> >
> >
> >
> > > Secondly, I somewhat understand the experimentation, but what I do not get is the significance of the results after reading the manuscript. I do not think that the authors have made it very clear to me and possibly, to the other reviewers too. I suggest that the authors should have a discussion section where they discuss the significance of their results. Also, for context, after reading lines 477-479, they did not give me that sense anyhow.
> >
> >
> > We appreciate your suggestions regarding the description of our experimental results. In response, we have made substantial revisions to better clarify the relationship between our experimental results and the theoretical aspects. We have moved the experimental results to the appendix and have added dedicated discussions in Section 5.3 and Appendix F to discuss the significance of our findings. These discussions are positioned below each corresponding experiment and link the experimental results to the theory. Regarding line 477-479, in that experiment, we compute the Euclidean distances between random sequences processed by a pre-trained GPT-2 model. We utilized the 'cdist' function from 'scipy.spatial.distance' to calculate the nearest neighbor distances and applied the 'min' function from 'numpy' to determine the nearest neighbor distance for each vector. We hope that these result in a more focused and clear presentation of our work.
> >
> >
> >
> > > Thirdly, in the Hopfield section of the related work section, the authors should use Millidge et al. (2022) and Saha et al. (2023) instead of Millidge et al. and Saha et al.. The same can be said in Remark 1. This is simply a stylistic comment.
> >
> > Thank you for pointing out the stylistic inconsistencies. We have corrected these citations.
> >
> > We appreciate your feedback, and we are open to any further suggestions you may have to improve the quality and clarity of our manuscript.

---

> > > ### Comment · Reviewer_zNBU · 2024-11-24
> > >
> > > Thank you to the authors for their hard work and detailed response. I have taken a look at the revised manuscript again. The structure and narrative of the paper are much better than before. Specifically, section 4 for the paper tells a better story than previous version. I will raise the overall score to 6 and modified the presentation score to 4.
> > >
> > > To justify the overall score, ***I think the experimentation and its significance are still lacking***. But the mathematical perspective, provided by the work, seems interesting and great to me. Perhaps, the authors can expand on them in the next work if this one is accepted.
> > >
> > > I wish the authors the best of luck!

---

> > > > ### Author Response · Authors · 2024-11-25
> > > >
> > > > Thank you for your constructive feedback and for acknowledging the improvements we have made to our paper. We appreciate the increased score and your recognition of our efforts. We take your comments on experimentation to heart and will take this into consideration for future work. Thank you again for your support and positive feedback.

---

### Official Review · Reviewer_KA1g · 2024-11-02

**Soundness:** 4
**Presentation:** 4
**Contribution:** 3
**Rating:** 8
**Confidence:** 3

**Summary:**

The paper investigates the scaling laws of transformer models, under the lenses of energy-based models, specifically, Hopfield networks. The authors expand on the classic work of Ramsauer on MCHNs, and show how hierarchical models can be described using a simpler energy function, based on the euclidean distance instead of the dot product, that implements a kind of nearest neighbor. Such an energy function is not local anymore (that is, it is not the sum of the energies of the single layers).

**Strengths:**

The analysis in well made, the problem tackled is an important one in the literature, and the work is well structured and clear. Furthermore, the connection with the nearest neighbour search is (to my knowledge) novel. The experimental evaluation is also well performed.

**Weaknesses:**

The section on cross entropy loss is a little less clear than the rest of the work: Maybe you could add another ‘proposition’ or a statement that summarizes the results of the second part of the section?

Scaling laws in the related works: N and D are never introduced.

“*It has been shown that the feed-forward layers can be interpreted as persistent memory vectors, and the weights of the attention and feed-forward sub-layers can be merged into a new all-attention layer without compromising the model’s performance*” This part is not well explained in the introduction in my opinion: could the authors dedicate to this a couple more sentences? Or, remove it and leave it as it is later in the paper?

The provided energy function is not ‘local’, as it is in other energy based models such as standard Hopfield networks, predictive coding, etc. This is quite interesting. However, I feel there is little discussion on the consequences of this. Why is this important? What does it tell us that is different from the previous literature?

**Questions:**

Using the euclidean distance inside an Hopfield networks: This has already been done, and it has shown good performance in simple image retrieval tasks [1]. Hence, interestingly your approach could be seen as using universal Hopfield networks to model the energy function of the transformer model.


[1] Millidge, Beren, et al. "Universal hopfield networks: A general framework for single-shot associative memory models." International Conference on Machine Learning. PMLR, 2022

---

> ### Author Response · Authors · 2024-11-19
>
> We thank the reviewer for their positive endorsement of our work. We have improved our work based on the suggestions made by the reviewer. The answers to the specific queries of the reviewer are provided below:
>
> > The section on cross entropy loss is a little less clear than the rest of the work: Maybe you could add another ‘proposition’ or a statement that summarizes the results of the second part of the section?
>
> We appreciate your suggestion. In response, we have introduced Proposition 5 (Lines 421-423) to summarize the findings from the second part of the section. This proposition essentially establishes that the optimal relationship between the model size $N$ and data size $D$ is given by $N \propto O(D^2)$ under idealized conditions where well-separated patterns are learned from the data and memorized by the model.
>
> > Scaling laws in the related works: N and D are never introduced.
>
> We appreciate your attention to detail. We have revised the manuscript to include the definitions of N and D in the related works (Line 108).
>
> > “It has been shown that the feed-forward layers can be interpreted as persistent memory vectors, and the weights of the attention and feed-forward sub-layers can be merged into a new all-attention layer without compromising the model’s performance” This part is not well explained in the introduction in my opinion: could the authors dedicate to this a couple more sentences? Or, remove it and leave it as it is later in the paper?
>
> Thank you for your valuable feedback regarding the clarity of our introduction. We have removed the mentioned sentence concerning the interpretation of feed-forward layers and the merging of weights into a new all-attention layer. We agree that introducing this complex concept early on might overshadow the primary focus of the introduction and could potentially confuse the reader.
>
> > The provided energy function is not ‘local’, as it is in other energy based models such as standard Hopfield networks, predictive coding, etc. This is quite interesting. However, I feel there is little discussion on the consequences of this. Why is this important? What does it tell us that is different from the previous literature?
>
> We appreciate the reviewer's keen observation regarding the non-local nature of the energy function in our model, which indeed differs from traditional energy-based models such as standard Hopfield networks and predictive coding. Your comment has sparked a crucial discussion on the implications of this distinction. While the existing literature, such as Hierarchical Associative Memory (Krotov, 2021), employs a system of differential equations to design a global energy function that can encompass feedback connections, our proposed model tackles the global energy specific to feedforward architectures. Predictive coding networks, as referenced in (Tang et al., 2023) and (Li et al., 2023), incorporate recurrent connections; however, their dynamics are focused on minimizing the total squared prediction errors, which results in a limited connection to the attention mechanism. By framing the information retrieval properties of the Transformer as a sequence of associative memories, our model seeks to establish correlations between model size and the memorization of training data, all within the context of statistical physics. We have integrated this discussion into our revised manuscript in the new paragraph "Relationship to Prior Work" in Lines 829-844.
>
> > Using the euclidean distance inside an Hopfield networks: This has already been done, and it has shown good performance in simple image retrieval tasks [1]. Hence, interestingly your approach could be seen as using universal Hopfield networks to model the energy function of the transformer model.
>
> We appreciate the reviewer's recommendation. We were previously unaware of the universal Hopfield network, which is a valuable addition to our literature review. Recognizing the connection, we can indeed consider our proposed energy function as a special case within the broader framework of universal Hopfield networks. To reflect this insight, we have expanded our "Related Work" section to incorporate this perspective (Lines 139-140) and modified "Our Contribution" (Lines 89-90) accordingly.

---

> > ### Comment · Reviewer_KA1g · 2024-11-26
> >
> > Thank you for your detailed rebuttal, I am happy that my minor feedback has been incorporated.

---

> > > ### Author Response · Authors · 2024-11-26
> > >
> > > Thank you for your support and constructive feedback. We are pleased to hear that our revisions have been well-received. Your input is greatly valued in helping us improve the quality of our work.

---

### Official Review · Reviewer_Xt6m · 2024-11-03

**Soundness:** 2
**Presentation:** 2
**Contribution:** 2
**Rating:** 5
**Confidence:** 4

**Summary:**

This paper introduces a theoretical framework for studying *causal* Transformer models and their scaling properties from the perspective of Associative Memories. The paper studies a "stacked" colelction of attractor energy functions to model a multi-layered Transformer where the argmin of one layer's energy is passed to the next layer. The next layer will further optimize its energy function in the local region around the output of the first layer, and so on. The authors validate their theoretical claims on vanilla transformers and GPT models.

**Strengths:**

(S1) **A novel theoretical analysis** of transformers. The paper describes a causal transformer as a sequence of energy functions (but these energy functions only model attention, see weaknesses) that serve as surrogates for optimizing a global energy function (eq. (6)) according to  the MM algorithm. This is an interesting perspective that could shed light on what the fundamental operation of a transformer could be.

**Weaknesses:**

(W1) **The equation for $E_{MCHN}^\beta(x)$ [L250] is incorrect (or at the very least undefined)**, since $M$ seems to be written as a linear operator over a vector of dimension $n=T_{\text{max}}d_{\text{emb}}$, which is not the attention operation of transformers because the `LogSumExp` in this equation is the summation index over the memorized examples instead of the sequence dimension... Could the authors please clarify this equation? If it is incorrect, the following propositions in the paper no longer hold.

(W2) **The equation for attention [L203] is incorrect**. $Q$, $K$, and $V$ should each have a sequence dimension, whereas no sequence dimension is mentioned in this equation. Is this intentional or a typo? Given my other reservations about the soundness of the analysis, I am inclined to believe that the authors misunderstand the attention operation of transformers.

(W3) **Assumption 1 is a BIG assumption**. Assumption 1 assumes that *every sample in the training set is memorized verbatim* as a vector of length $n$, which is not a reasonable assumption for an LLM or even any Associative Memory trained on abundant data. Can the authors discuss the implications of this assumptions not holding in real-world scenarios? Or provide some empirical evidence/theoretical justification for why their results might still be relevant despite this assumptions?

(W4) **Eq (5) shows optimal $x^{(t)}$ being passed to the next layer $t+1$**. However, the update rule of a vanilla Transformer is not guaranteed to minimize the energy $E_{t}(x)$ and I wonder how the authors could have made the assumption that each step returns the `argmin` of the energy function for a particular layer. Vanilla transformers (using the config specified in App. F.1) are allowed to learn any value matrix in the attention, which could lead to the defined energy *increasing* after each update and not satisfy the assumption of Eq. (5)

(W5) **Unconvincing (and arguably unrelated) empirical results**. This is a fundamental weakness of the paper. The empirical results study vanilla Transformer architectures: architectures that include layer normalization and feed-forward networks, do not use the proposed metric in $g(x)$ from Eq. (2), whose update rule is not guaranteed to minimize the energy function on L250, and whose attention operation is not reasonably modeled by Eq. (4). I don't know how to reconcile this weakness. Given the fundamental incompatibility with the experimental results and the actual Transformer architecture, I am surprised that the theoretical results "align closely" [L449] with the empirical results and would ask the authors to dig deeper into why this could be the case.

(W5) **Overloaded notations make exposition unclear**. The mathematical notation $d$ is a little confusing. It represents:

1. The number of samples in the training set or validation set [L148-149]
2. The embedding dimension of each token [L152]
3. The metric [L156]

**Summary**

As mentioned in the Strengths section above, I find the perspective of MM as a description of what the Transformer architecture is doing fascinating. I am willing to reconsider my score if the authors can clarify where I have fundamentally misunderstood the paper. I cannot recommend this paper in its current state for acceptance.

**Questions:**

(Q1) Why bother making a distinctive definition of a validation set at all? If $\tilde{\mathcal{D}} \subset \mathcal{D}$ [L191] i.e., the validation set is some subset of the memorized training points, why consider it at all?

(Q2) The analysis of sections 4 and 5 seems to have nothing to do with the actual transformer architecture.

(Q3) [L348-352] The attention softmax is irrelevant to the final cross-entropy, and including this statement here only confused me. Can the authors justify why we care about the softmax in the attention when considering the softmax in the cross-entropy loss?

**(More) Typos:**
- Eq (3) has incorrect indices of summation, if it is supposed to align with $E_{\text{MCHN}}$.
- L252: Typo, the definition of LogSumExp RHS uses $\rho^{i}$, when the function is defined in terms of $Mx$.

**Improvements**
(I1) Scaling laws (eq (1)) $N$ and $D$ are not defined. Is $N$ the same as the definition in line 153? Is $D$ the same as the definition in line 151? If so, please put these definitions closer to when they are first introduced

(I2) [L122] A bit non-standard notation for the partition function $Z_{\theta}$ -- generally this is expressed with integrating variable `dx` on the RHS

**Additional Comments**
(C1) The following statement is deceptive:

> As the attention layers and the FF layers contribute to the majority of the model’s parameters...

This is only true when we ignore the embedding and unembedding layers of a Transformer, which can be incredibly large with large vocab sizes. For smaller transformer models with large vocab sizes, it is possible that the embedding and unembedding matrices dominate the parameter count.
Take for example `Model 1` in App F.1, where the number of parameters in the embedding+unembedding layers given the vocab size is ~96M parameters, whereas the reported total number of transformer params is 40M.
Please clarify, perhaps "The MHA and FF layers account for most of the parameters outside the embedding/unembedding layers" or "The fundamental operations of the Transformer are the MHA and FF layers", since parameter count is not the reason why you are focusing on these operations.

(C2) A gripe: the authors treat their version of the transformer as "standard", when it is arguably wrong or at least far from standard. This is summarized in the following quote from L419-420

> Also, modifications to the transformer blocks, such as additional layer normalization may contribute to the lower bound of the cross-entropy.

It is an open challenge to get high-performant transformers without normalization, and all modern transformers (including the models tested in this work!) use some form of normalization.

---

> ### Author Response · Authors · 2024-11-19
>
> Thank you for your time and suggestions. We address each of your comments below.
> * (W1) We sincerely appreciate the reviewer's observation regarding the inconsistency in our notation. We acknowledge that we adopted the MCHN definition from Ramsauer et al. (2020) without thoroughly aligning the notations. In our revised manuscript, we have included the definition of $M$ and replaced $x$ with the boldface $\mathbf{x}$ to clearly denote the vector form. The following theoretical results remain unaffected by these notational discrepancies.
> * (W2) We apologize for any confusion caused by our previous notation. We would like to clarify that the matrices denoted by $Q, K$ and $V$ are indeed weight matrices. To avoid any ambiguity, we have revised the introduction of our attention mechanism section by replacing these notations with $W_Q, W_K$ and $W_V$ respectively. Again, these do not affect the following theoretical results.
> * (W3) We appreciate the reviewer's question. For a comprehensive response addressing the assumptions, please refer to the global response.
> * (W4) In Propositions 1, 2, and 3, we have established that both the proposed energy and the MCHN energy, previously demonstrated to be closely related to the attention mechanism, can be effectively approximated through an argmin search. Moreover, the MCHN update has been shown to converge in a single step. Given our focus on the collective behavior of attention layers, we consider the current approximation to be reasonable.
> * (W5) Although our paper primarily focuses on the theoretical analysis of the transformer's behavior, rather than an empirical examination of scaling laws, we recognize the constraints of our current experimental setup. Regarding the theoretical results in [L449], please refer to our detailed responses on empirical results in the global response.
> * (W6) Thank you for the suggestion. To improve the clarity, we have added a summary of notations in Appendix A.
> * (Q1) We appreciate the reviewer's inquiry. Initially, we differentiated between the two sets in Assumption 2, which stated that held-out samples are stored in a manner analogous to the training set. Our revision now articulates Assumption 2 to emphasize the consistency of latent patterns. We invite the reviewer to review our responses regarding the assumptions in the global rebuttal.
> * (Q2) The paper is structured into two parts. The earlier sections, including Section 4, are dedicated to presenting our model and introducing a novel energy function for the transformer-based networks using associative memory. Section 5, in particular, builds upon this foundation by applying our model to analyze the cross-entropy loss within the context of the pre-training. This analysis not only validates the theoretical framework but also elucidates its practical relevance to the transformer model.
> * (Q3) Thank you for your question. It appears there may be a misunderstanding regarding the role of the softmax function in both the attention mechanism and the CE loss within the training process. To clarify, within the attention mechanism, the softmax function ensures that the model assigns appropriate importance to different elements within the input sequence. In the final layer of the transformer before calculating the CE loss, the softmax function converts the output logits from the transformer into a probability distribution, which represents the model's predictions for the next token in the sequence. The CE loss then measures the difference between these predicted probabilities and the actual target distribution, providing a measure of how well the model's predictions match the true labels. The attention softmax influences the model's ability to understand and process the input data, which in turn affects the output probabilities that are used to calculate the CE loss. In essence, the attention softmax indirectly influences the loss by shaping the model's predictions. We have made revisions to the manuscript to ensure that this relationship is more clearly articulated.
> * Typos and Improvements: Thank you. We have corrected the identified typos and made revisions to the manuscript in accordance with your suggestions (I1) and (I2) as requested.
> * (C1) Thank you. Our intuition is that as the model scales, the attention and FF layers, being stacked, constitute the majority of the model's parameters. However, taking your suggestions into consideration, we have further clarified that the fundamental operations of the Transformer are indeed the attention and FF layers.
> * (C2) We appreciate the reviewer's feedback. As discussed in the paper (L406-408), the stability of the probabilities is attributed to the application of normalization operators which regulate the distribution of activations. In response to the specific point raised about our description at lines L419-420, our intention was to emphasize that our exploration is narrow in scope without considering other potential modifications.

---

> > ### Comment · Reviewer_Xt6m · 2024-11-22
> >
> > I thank the authors for their response and clarifications. After reading all the other reviews and responses, I remain unconvinced by the methodology, experiments, and presentation of this paper.
> >
> > 1. **Misinterpretation of MCHN Energy in the context of Transformers**
> > The MCHN energy of Ramsauer et al. is employed as a **token lookup mechanism**, not as a lookup over entire sequences as assumed in this work (Assumption 1). A query corresponds to a single token, while stored patterns relate to other tokens within the same sequence. This distinction is critical as it impacts the motivations of the theoretical framework and is the entire reason that Transformers are able to perform so much better than a naive DAM on language since it is much easier to memorize *token relationships* than *entire sequences*.
> >
> > 2. **Empirical Results have little relevance to theoretical results**
> > I am of the same [opinion as Reviewer zNBU](https://openreview.net/forum?id=tJE9WeqHEI&noteId=MMd2z3Dwf5): I am concerned about the significance and meaning of the empirical results. The results appear disconnected from the theoretical model being analyzed. Specifically,
> >
> >     - The nearest neighbor distances are plotted based on a sampling of OpenWebText (OWT) and predicted tokens, not predicted sequences, as the theory suggests.
> > 	- The discussion around Proposition 4 and its experimental verification (Fig 2 bottom left) remains unclear. Were models explicitly trained with a total number of parameters $N \sim O(D^2)$? If not, the connection between theory and experiments is tenuous -- how else could you achieve the lower bounded loss = 1.?
> >    - The overall experimental setup seems designed to study vanilla Transformers rather than to validate the novel energy functions introduced in the paper.
> >
> > I believe additional discussion is necessary to clarify the approach.
> >
> > 3. **Proposition 5 and Model Size Complexity do not align with behavior of real Transformers**
> > Proposition 5 posits that the optimal number of parameters $N$ for a Transformer should scale quadratically with the dataset size $D$ for a well-separated dataset. This claim contradicts the widely accepted understanding that Transformers learn compressed representations of data. If $N = O(D^2)$, the proposed balance between model and data sizes seems impractical and misaligned with existing theoretical and empirical evidence.
> >
> >
> > Having read this paper several times over and carefully considering its claims, I agree that this work explores an interesting theoretical direction. However, the gap between the theoretical framework and the experimental setup weakens the paper’s contribution, and several claims (e.g., Proposition 5, Assumption 1) contradict established understanding of how Transformers operate.
> >
> > I appreciate the effort the authors put into this work and acknowledge that other reviewers have found value in its contributions. However, after reading this paper's claims over several times, I cannot in good conscience raise my score. I defer to the area chair to make the final call regarding the acceptance of this work.

---

> > > ### Author Response · Authors · 2024-11-23
> > >
> > > We appreciate the reviewer's continuous engagement in the discussion.
> > >
> > > > 1. The MCHN energy of Ramsauer et al. is employed as a token lookup mechanism, not as a lookup over entire sequences as assumed in this work (Assumption 1). A query corresponds to a single token, while stored patterns relate to other tokens within the same sequence. This distinction is critical as it impacts the motivations of the theoretical framework and is the entire reason that Transformers are able to perform so much better than a naive DAM on language since it is much easier to memorize token relationships than entire sequences.
> > >
> > > Thank you for your question. It appears that there may be a misunderstanding regarding the application of the MCHN energy. To clarify, the energy functions presented in our paper, including the layer-wise energy in Eq. (4) and the global energy in Eq. (7), consider patterns as n-dimensional latent representations rather than individual tokens. A query then corresponds to a perturbed pattern which can indeed be retrieved using update mechanisms similar to those described in (Ramsauer et al., 2020). This approach allows us to perform a "lookup" not on a single token, but on the relationships between tokens within a sequence. We hope this clarification addresses your question and provides a better understanding of how the MCHN energy relates to our model. If further clarification is needed, we are more than happy to provide additional details.
> > >
> > >
> > > > 2. I am of the same opinion as Reviewer zNBU: I am concerned about the significance and meaning of the empirical results. The results appear disconnected from the theoretical model being analyzed. Specifically,
> > > >- The nearest neighbor distances are plotted based on a sampling of OpenWebText (OWT) and predicted tokens, not predicted sequences, as the theory suggests.
> > > >- The discussion around Proposition 4 and its experimental verification (Fig 2 bottom left) remains unclear. Were models explicitly trained with a total number of parameters $N \sim O(D^2)$? If not, the connection between theory and experiments is tenuous -- how else could you achieve the lower bounded loss = 1.?
> > > >- The overall experimental setup seems designed to study vanilla Transformers rather than to validate the novel energy functions introduced in the paper.
> > >
> > > Thank you for your questions. We have made significant revisions to our paper to clarify the connection between our experiments and the theoretical aspects of our work, as detailed in our response to all reviewers above. Below, we provide additional information to address your specific questions:
> > >
> > > - In our first experiment, we randomly sampled 100K chunks, each containing $T_{max}=256$ tokens, from the OpenWebText dataset and recorded the activation vectors from the $l$-th layer. These vectors represent the latent patterns, which are the n-dimensional representations alluded to in the theoretical model, rather than individual tokens.
> > >
> > > - In our second experiment, we trained vanilla Transformers with varying parameters using a fixed dataset. The dataset's limited vocabulary ensures the well-separated condition in Assumption 3. The primary goal of this experiment was to investigate the lower bound of the CE loss, which is central to Proposition 4, rather than to directly study the $N \sim O(D^2)$ relationship. Our results align with Proposition 4.
> > >
> > >
> > > - While our overall experimental setup includes studying vanilla Transformers, it is also designed to validate the novel energy functions introduced in our paper. We conduct a series of experiments to evaluate the radius with GPT-2, train vanilla Transformers, and train models with varying widths and dimensions, which correspond to validating Equation 14, Proposition 4, and Proposition 5, respectively. More detailed descriptions regarding the objectives and significance of the experiments can be found in Section 5.3 and Appendix F.

---

> > > > ### Author Response · Authors · 2024-11-23
> > > >
> > > > > 3. Proposition 5 posits that the optimal number of parameters $N$ for a Transformer should scale quadratically with the dataset size $D$ for a well-separated dataset. This claim contradicts the widely accepted understanding that Transformers learn compressed representations of data. If $N = O(D^2)$, the proposed balance between model and data sizes seems impractical and misaligned with existing theoretical and empirical evidence.
> > > >
> > > > As has been mentioned in the related work, the relationship between our results and empirical scaling laws is discussed in Appendix A. To recap here, in our experiments, we utilize a reduced dataset to emulate the conditions of pattern separation and memorization. Moreover, we have trained the models to ensure that the training losses have stabilized and the test losses have begun to ascend, indicative of a mild degree of over-parameterization. These conditions diverge from those of the Chinchilla experiment. In practical scenarios, commercial LLMs, akin to the Chinchilla model, are not subjected to such conditions. As we have noted in our paper, it has been observed that even after training on up to 2T tokens, some models have yet to exhibit signs of saturation. Therefore, our current experimental setup represents an idealized condition that has not been encountered in commercial LLMs.
> > > >
> > > > > Having read this paper several times over and carefully considering its claims, I agree that this work explores an interesting theoretical direction. However, the gap between the theoretical framework and the experimental setup weakens the paper’s contribution, and several claims (e.g., Proposition 5, Assumption 1) contradict established understanding of how Transformers operate.
> > > >
> > > > Thank you for acknowledging the theoretical interest the paper presents. We hope that our responses have alleviated any doubts you may have had regarding Proposition 5 and Assumption 1. If additional clarification is needed, we are more than willing to provide further explanations.

---

> > > > ### Comment · Reviewer_Xt6m · 2024-11-25
> > > >
> > > > > ... this approach allows us to perform a "lookup" not on a single token, but on the relationships between tokens within a sequence.
> > > >
> > > > When each pattern stored in an MCHN consists of a concatenated sequence of tokens, and you compare a new pattern to each stored pattern, you are performing a naive "sequence similarity" between the two patterns. In order to learn the "relationships between tokens", you need to have a mechanism to allow tokens to talk to other tokens in the same sequence. The MCHN in this paper does not have such a mechanism, whereas traditional attention does. For instance, consider a sequence that consists of token ids $s_{A}:=[0, 1, \ldots, n]$ and another sequence $s_{B} := [n, 0, 1, \ldots, n-1]$. $s_{A}$ and $s_{B}$ have the same tokens, but the positions of these tokens are offset by 1. In the MCHN considered in this work, these two sequences would be considered vastly different (i.e., they would have a very high distance from each other). This is not memorizing token relationships -- this is memorizing strict sequences.
> > > >
> > > >
> > > > I thank the authors for clarifying the other aspects of this paper. The model studied in this work is not the Transformer by any means, but I have been too harsh in my assessment of the paper's soundness and presentation. I will slightly raise my score.

---

> > > > > ### Author Response · Authors · 2024-11-26
> > > > >
> > > > > Thank you for your willingness to reconsider your assessment, and we value your support in raising your score.
> > > > >
> > > > > > When each pattern stored in an MCHN consists of a concatenated sequence of tokens, and you compare a new pattern to each stored pattern, you are performing a naive "sequence similarity" between the two patterns. In order to learn the "relationships between tokens", you need to have a mechanism to allow tokens to talk to other tokens in the same sequence. The MCHN in this paper does not have such a mechanism, whereas traditional attention does.
> > > > >
> > > > > Thank you for your question. It appears that there may be a misunderstanding. As demonstrated in [Ramsauer et al. (2020)](https://openreview.net/pdf?id=tL89RnzIiCd), the MCHN dynamic is similar to the attention mechanism, and our paper does not propose the MCHN energy function. When the reviewer refers to "comparing a new pattern to each stored pattern," we interpret this as the MCHN dynamics retrieving a stored pattern that closely matches the query pattern using a dynamic akin to the attention mechanism as induced by their energy function. In this context, the tokens within a sequence do indeed "talk to other tokens", similar to the attention mechanism.
> > > > >
> > > > >
> > > > > > For instance, consider a sequence that consists of token ids $s_{A}:=[0, 1, \ldots, n]$ and another sequence $s_{B} := [n, 0, 1, \ldots, n-1]$. $s_{A}$ and $s_{B}$ have the same tokens, but the positions of these tokens are offset by 1. In the MCHN considered in this work, these two sequences would be considered vastly different (i.e., they would have a very high distance from each other). This is not memorizing token relationships -- this is memorizing strict sequences.
> > > > >
> > > > >
> > > > >
> > > > > We would like to reiterate that this paper does not introduce the MCHN energy. In your example, you represent the sequences with token IDs rather than embeddings. Nonetheless, to elucidate, within the MCHN framework, if we consider $s_A$ as a stored pattern and $s_B$ as a query pattern in the latent space, then their Euclidean distance is $\sqrt{n(n+1)},$ and their cosine similarity is $\cos \alpha = \sqrt{2(n-1)/(2n+1)}.$
> > > > > Using notations from (Ramsauer et al., 2020), Eq. (308): $M=\sqrt{n(n+1)(2n+1)/6},$
> > > > > $\Delta = M^2(1-\cos \alpha) = n(n+1)/3.$
> > > > > According to Eq. (300) of Ramsauer et al. (2020), the MCHN dynamics will work as long as the number of patterns $N$ satisfies
> > > > > $n(n+1)/3\geq 2/N + \ln (N(N-1)n(n+1)(2n+1)/3)$,
> > > > > This condition can be met, for instance, by setting $n=6$ and $N=80$.
> > > > > Therefore, the MCHN dynamics can effectively capture the relationships between tokens in sequences.
> > > > >
> > > > > We hope that our responses have cleared your concerns. If additional clarification is needed, we are more than willing to provide further explanations.

---

### Author Response · Authors · 2024-11-19
**Comments for All Reviewers**

We greatly appreciate the time, effort, and suggestions provided by the reviewers. Your feedback not only helps us improve the paper but also encourages us to think more deeply about our work. We extend our special thanks to reviewers KA1g, zNBU, and JTCX for recognizing the value of our mathematical analysis. We will begin by addressing some common queries and concerns. We have updated the manuscript to reflect all the revisions below.


## Regarding the assumptions

In response to the valuable feedback from Reviewers Xt6m and JTCX, we recognize the limitations in our assumptions. Our initial phrasing might have implied a verbatim memorization of samples within dataset $\mathcal{D}$ as patterns to be encoded by the network. Upon reflection, we understand that this could potentially rely on the separation of patterns. In light of the insightful suggestion from Reviewer zNBU, we have revised **Assumption 1** and **Assumption 3** to provide greater clarity. We now clarify (starting from Line 170) that the Transformer blocks are responsible for storing the latent patterns derived from the text data after it has been passed through an embedding layer, which inherently reduces the correlation between the original sequences.
Regarding **Assumption 2**, we have accordingly revised the original assumption to one of latent representation consistency, i.e., we assume that the latent representations derived from the validation set, after being processed through an embedding layer, are stored in a manner analogous to those extracted from the training set. These refinements have been integrated into our revised manuscript. Given that certain idealized assumptions are essential for theoretical exploration, we believe that the proposed model still reveals significant phenomena pertaining to Transformer performance.


## Additional empirical results
In response to the valuable feedback from Reviewers Xt6m, zNBU, and JTCX, we have expanded our experimental scope. We have managed to increase the collected activation from 1\% to 1.3\%, and the corresponding discussions have been updated in our revision. While computational constraints currently limit the scale of our experiments, this increment in data size enables a more robust verification of the radius value. As can be seen from Fig. 2, our new radius estimations are similar to the previous ones. This additional analysis provides further confidence in the consistency of our methodology.



We sincerely appreciate the reviewers' thorough evaluation and constructive feedback. We have updated the manuscript to reflect all the revisions. To foster ongoing discussion, we warmly invite the reviewers to continue the dialogue and confirm whether their concerns have been addressed.

---

> ### Author Response · Authors · 2024-11-23
> **Comments for All Reviewers**
>
> Once again, we thank the reviewers for continuing to provide valuable suggestions. In response to the concerns raised by Reviewers Xt6m and zNBU regarding the significance of the experimental results in our theory-focused paper, we have made substantial revisions. Specifically, we have relocated all experimental sections to Appendix F, where we have also included additional descriptions of their significance. This change allows us to allocate more space in the main text for the mathematical derivations that are central to our study. In the main content, we now offer a concise summary of the experiments and their significance in Section 5.3. Consequently, we have removed all summary statements related to the experimental contributions that were previously present in the abstract, "our contribution", and the conclusion.
>
> This revision emphasizes the theoretical findings of our research. However, we remain open to the reviewers' preferences. If they believe that retaining the empirical results within the main body of the paper is more appropriate, we are willing to revert the changes and incorporate those sections back into the main text.

---

### Meta-Review · Area_Chair_hV1d · 2024-12-20

**Metareview:**

The authors present a mostly theoretical analysis of how performance in Transformers scales with model size, based on an approximation of Transformers as a sequence of energy-based models. The reviewers agreed the analysis was interesting and novel, but two of the reviewers took issue with the empirical results in the paper, having concerns that the analysis did not support the conclusions of the theoretical framework. In response to this, the authors simply moved the empirical results to the appendix instead of improving the presentation or providing new results. Given the paper is not a theory paper in the sense of proving some novel theorem, but is meant to be a useful approximation to real networks, some empirical justification for the simplifying assumptions is necessary. Therefore I recommend against acceptance.

**Additional Comments On Reviewer Discussion:**

This was mainly discussed in the metareview already.

---

### Decision · Program_Chairs · 2025-01-22

Reject